Host-Microbe Biology
# Correlations between the Composition of the Bovine Microbiota and Vitamin B$_{12}$ Abundance

Julian Franco-Lopez,[a] Melissa Duplessis,[b] An Bui,[a] Coralie Reymond,[c] William Poisson,[d] Lya Blais,[e] Jasmine Chong,[f] Rachel Gervais,[d] Daniel E. Rico,[d,g] Roger I. Cue,[c] Christiane L. Girard,[b] Jennifer Ronholm[a,c]

[a]Department of Food Science and Agricultural Chemistry, Faculty of Agricultural and Environmental Sciences, Macdonald Campus, McGill University, Montreal, Quebec, Canada

[b]Agriculture et Agroalimentaire Canada, Centre de Recherche et Développement de Sherbrooke, Sherbrooke, Quebec, Canada

[c]Department of Animal Science, Faculty of Agricultural and Environmental Sciences, Macdonald Campus, McGill University, Montreal, Quebec, Canada

[d]Département des Sciences Animales, Université Laval, Québec, Quebec, Canada

[e]Département de Microbiologie et d'Infectiologie, Université de Sherbrooke, Sherbrooke, Quebec, Canada

[f]Institute of Parasitology, Faculty of Agricultural and Environmental Sciences, Macdonald Campus, McGill University, Montreal, Quebec, Canada

[g]Centre de Recherche en Sciences Animales de Deschambault, Deschambault, Quebec, Canada

**ABSTRACT** Vitamin B$_{12}$ is synthesized by prokaryotes in the rumens of dairy cows—and this has implications in human nutrition since humans rely on consumption of dairy products for vitamin B$_{12}$ acquisition. However, the concentration of vitamin B$_{12}$ in milk is highly variable, and there is interest in determining what causes vitamin B$_{12}$ variability. We collected 92 temporally linked rumen, fecal, blood, and milk sample sets from Holstein cows at various stages of lactation fitted with rumen cannula and attempted to define which bacterial genera correlated well with vitamin B$_{12}$ abundance. The level of vitamin B$_{12}$ present in each sample was measured, and the bacterial population of each rumen, fecal, and milk sample ($n = 263$) was analyzed by 16S rRNA-targeted amplicon sequencing of the V4 region. The bacterial populations present in the rumen, small intestine, and milk were highly dissimilar. Combined diet and lactation status had significant effects on the composition of the microbiota in the rumen and in the feces. A high ruminal concentration of vitamin B$_{12}$ was correlated with the increased abundance of *Prevotella*, while a low ruminal concentration of vitamin B$_{12}$ was correlated with increased abundance of *Bacteroidetes, Ruminiclostridium,* and *Butyrivibrio*. The ultimate concentration of vitamin B$_{12}$ is controlled by the complex interaction of several factors, including the composition of the microbiota. Bacterial consumption of vitamin B$_{12}$ in the rumen may be more important in determining overall levels than bacterial production.

**IMPORTANCE** In this paper, we examined the microbiome of the bovine rumen, feces, and milk and attempted to understand how the bacterial communities at each site affected the production and movement of vitamin B$_{12}$ around the animal's body. It was determined that the composition of the bovine rumen microbiome correlates well with vitamin B$_{12}$ concentration, indicating that the rumen microbiota may be a good target for manipulation to improve production of this important vitamin.

**KEYWORDS** bovine, microbiome, microbiota, milk, rumen, vitamin B$_{12}$

Vitamin B$_{12}$ is an essential nutrient in human nutrition that can be synthesized only by bacteria and archaea in the presence of an adequate cobalt supply (1). Humans generally acquire vitamin B$_{12}$ through the consumption of animal products, especially ruminant products, such as beef or dairy products, because it is synthesized by bacteria and archaea in the rumens of cattle (2, 3). Dairy products are an ideal source of vitamin

This article followed an open peer review process. The review history can be read here.

Address correspondence to Jennifer Ronholm, Jennifer.Ronholm@mcgill.ca.

The composition of the bovine rumen microbiota plays an important role in determining the abundance of vitamin B12 in the milk we drink!

$B_{12}$ since the molecule is very stable, can survive pasteurization, and is not destroyed by daylight or prolonged refrigerator storage (4). Moreover, it has been shown that vitamin $B_{12}$ from milk is more bioavailable than the synthetic form found in supplements (5). However, the concentration of vitamin $B_{12}$ found in milk varies considerably among cows and herds. According to the U.S. Department of Agriculture (2012), 250 ml of milk should provide 46% of the 2.4-$\mu$g recommended dietary allowance of vitamin $B_{12}$ required for a human above 13 years old (6). However, at the herd level, a 250-ml glass of milk can contain 0.575 to 1.473 $\mu$g of vitamin $B_{12}$, and this variability increases to 0.17 to 5.394 $\mu$g if the milk is taken from individual dairy cows (7).

After synthesis in the rumen, vitamin $B_{12}$ is absorbed in the ileum in cows (8). In the small intestine, prior to reaching the ileum, vitamin $B_{12}$ is bound by intrinsic factor (IF), produced by the parietal cells of the stomach, and the IF-vitamin $B_{12}$ complex is bound by a receptor that is expressed on enterocytes in the ileum and absorbed into the enterocyte (8, 9). Inside the enterocyte, IF is degraded, and vitamin $B_{12}$ is released into circulation in blood from the basolateral side (10). In blood, transcobalamin binds vitamin $B_{12}$ and is responsible for trafficking the vitamin to the tissues (9). In dairy cows, 46% of the vitamin released from the gastrointestinal tract is removed by the liver (8). In humans, mammary epithelial cells have a high affinity for transcobalamin-bound vitamin $B_{12}$ (11, 12). The transmembrane receptor CD320 is expressed on mammary epithelial cells and shows a high affinity for transcobalamin-bound vitamin $B_{12}$ (12). Upon endocytosis by mammary epithelial cells, transcobalamin is degraded in the cell, and free vitamin is transported into milk (13). The same transport proteins that are active in this process in humans have also been found in cattle, and therefore, the process is thought to be similar between these species (14). In dairy cows, uptake of vitamin $B_{12}$ by the mammary gland, although closely related to the plasma vitamin concentration in the mammary artery, represents only 5.5% of the concentration of the vitamin in plasma. Moreover, uptake of the vitamin by the mammary gland is 17% greater than the amount secreted in milk (15).

Synthesis of vitamin $B_{12}$ in the rumen of the cow is influenced by diet composition. Since cobalt is at the core of the vitamin $B_{12}$ molecule, ruminal synthesis of the vitamin requires an adequate dietary cobalt supply (1). Moreover, increasing the forage-to-concentrate ratio in the diet increases the concentration of vitamin $B_{12}$ in solid- and liquid-associated bacteria in the rumen (16). Apparent synthesis of vitamin $B_{12}$ in the rumen is positively associated with the dietary concentrations of neutral detergent fiber (NDF) and acid detergent fiber (ADF) but negatively correlated with dietary starch concentration (17–19). Indeed, apparent synthesis of vitamin $B_{12}$ in the rumen is threefold greater in cows receiving a high-fiber diet than in cows receiving a high-starch diet (18). Genetic selection appears to also play a role in the variability of vitamin $B_{12}$ concentrations in milk, and the heritability value has varied from 0.23 to 0.37 in different studies (14, 20), although the extent to which this analysis is confounded by microbiome composition has not been addressed. Neither feed differences nor bovine genetics fully explain the variation of vitamin $B_{12}$ in milk (7, 14, 20).

The microbial synthesis of vitamin $B_{12}$ is complex due to the elaborate structure and high metabolic cost of its synthesis. Only a very few, but phylogenetically diverse, bacteria and archaea are known to produce it (21, 22). Vitamin $B_{12}$ can be synthesized via a *de novo* or a salvage pathway, and both pathways can occur under either anaerobic or aerobic conditions (23). In the *de novo* pathway, bacteria synthesize vitamin $B_{12}$ from glutamate and cobalt; this process is complex and involves the activity of more than 30 bacterial genes (1). The salvage pathway uses ATP hydrolysis for the uptake of other existing corrinoids and their conversion into vitamin $B_{12}$ (21). Several bacterial and archaeal lineages carry genes that encode enzymes that perform radical rearrangements or methyl transfer reactions, which require vitamin $B_{12}$ as a cofactor (22). Like humans and other eukaryotes, most bacteria require vitamin $B_{12}$ as a cofactor for methylmalonyl coenzyme A (CoA) mutase (MCM) and corrinoid-dependent methionine synthase (MetH); the former catalyzes the interconversion of *R*-methylmalonyl-CoA and succinyl-CoA, which is a critical step in the metabolism of amino acids, fatty

acids, cholesterol, and sugar to propionate (24), while the latter catalyzes the final step in the biosynthesis of methionine (22). Unlike eukaryotes, some bacterial species can also use MCM to convert succinyl-CoA to methylmalonyl-CoA, a process that ultimately generates $CO_2$ and has been hypothesized to facilitate respiration in anaerobic environments, like the rumen or intestine (25). Vitamin B$_{12}$ is also required as a cofactor for ethanolamine ammonia lyase which allows bacteria to use ethanolamine, which is commonly found in the gut, as a source of carbon and nitrogen (26). Since several prokaryotes that require vitamin B$_{12}$ for survival also lack the ability to synthesize vitamin B$_{12}$, they must acquire it from other prokaryotes in their environment. Although not known for dairy cows, approximately 80% of the commensals that comprise the human intestinal microbiota appear to require exogenous vitamin B$_{12}$ based on the presence of transport relevant transport systems, while less than 25% of human commensals can produce the vitamin through *de novo* synthesis (22).

Since the variation of vitamin B$_{12}$ in milk is not fully explained by herd management practices, diet, or heritability, and vitamin B$_{12}$ is synthesized only by bacteria and archaea, it is likely that the composition of bovine microbiota plays a role in the observed variability. In this exploratory study, we have attempted to identify specific bacterial genera that are correlated with the abundance of vitamin B$_{12}$ in the bovine rumen, feces, and milk for the purpose of targeting future experimental work in the field.

## RESULTS

In this investigation, 92 rumen, blood, and fecal samples, and 71 milk samples were collected from 50 Holstein dairy cows fitted with rumen cannula. Sampling was conducted over a 6-month period, and individual animals provided two entire sample sets (rumen, blood, fecal, and milk) at two different time points approximately 3 months apart. Animals that were culled or that experienced a mastitis infection requiring antibiotic treatment were eliminated from the study and may have contributed only one sample set. Lactating animals were fed one of two similar lactation diets, group 1 ($n = 53$) or group 2 ($n = 21$) for the duration of the study, and animals that finished lactation during sampling were fed a close-up diet ($n = 7$), followed by a dry diet ($n = 11$) (Table 1). Milk samples were not collected from nonlactating animals, but fecal, blood, and rumen samples were collected from dry animals for the purposes of comparison. The microbiome of each rumen, fecal, and milk sample was characterized via 16S rRNA-targeted amplicon sequencing, and the vitamin B$_{12}$ concentration of each sample was measured. The data sets were combined, and bacterial genera that were correlated with vitamin B$_{12}$ concentration were identified.

**The composition of the bovine microbiome is niche specific.** After quality filtering of the raw sequencing data and removing samples that had <10,000 high-quality paired-end sequences, bovine fecal samples ($n = 92$) and corresponding rumen ($n = 88$) and milk samples ($n = 71$) were kept for downstream statistical analysis. The sequencing output for both the extraction and sequencing of negative controls was low, not indicative of contamination, and therefore not considered in any downstream analysis. After rarifying to 10,953 sequences, clustering into operational taxonomic units (OTUs) at the 97% similarity threshold resulted in the identification of 5,028 unique OTUs that were observed a minimum of 20 times in the samples. Analysis of the alpha-diversity in each niche revealed that both taxonomic richness (Chao1 $P < 0.001$; Kruskal-Wallis statistic, 168.8) and diversity (Shannon $P < 0.001$; Kruskal-Wallis statistic, 65.8) were significantly different between niches (Fig. 1A and B). In terms of beta-diversity, nonmetric dimensional scaling (NMDS), analysis of similarity (ANOSIM), and permutational multivariate analysis of variance (PERMANOVA) analysis of Bray-Curtis dissimilarities revealed that the microbiota of each niche was highly dissimilar (ANOSIM, $R = 0.96$, $P < 0.001$; PERMANOVA, $F = 160.57$, $P < 0.001$). In addition, a test for homogeneity of multivariate dispersion (PERMDISP) was used to indicate that the population variances within each niche were also significantly different (PERMDISP, $P < 0.001$, F value, 148.45) (Fig. 1C). Each of the characterized niches was dominated by the

**TABLE 1** Analysis of diets fed to the four groups

| Ingredient[a] | % composition or nutrient composition of diet fed to the groups[b] | | | |
|---|---|---|---|---|
| | Dry[c] | Close-up[c] | Group 1[d] | Group 2[d] |
| **Ingredients** | | | | |
| Hay | 35.8 | | | |
| Grass hay | 33.5 | 36.8 | 2.6 | 2.0 |
| Legume-grass silage | | | 23.0 | 39.2 |
| Corn silage | 11.5 | 34.0 | 34.1 | 26.1 |
| Cracked corn | | | 16.4 | 15.1 |
| Soybean meal | 13.4 | 16.3 | 8.4 | 9.2 |
| Beet pulp | 3.5 | 10.0 | 3.5 | |
| Mineral and vitamin premix | 1.4 | 1.4 | 1.5 | 1.4 |
| Calcium carbonate | 0.5 | 1.2 | 1.0 | 1.0 |
| Distiller grain (corn) | | | 2.6 | 1.8 |
| Corn gluten meal | | | 2.6 | 1.8 |
| Canola meal | | | 1.7 | 1.2 |
| Micronized soybean | | | 1.7 | 1.2 |
| Urea | 0.39 | 0.3 | | |
| Megalac | | | 0.97 | |
| | | | | |
| **Nutrient composition (% of DM unless otherwise specified)** | | | | |
| DM | 58.0 | 44.7 | 43.3 | 42.2 |
| CP | 14.4 | 14.6 | 15.0 | 15.2 |
| ADF | 28.5 | 22.9 | 16.1 | 19.0 |
| NDF | 50.7 | 42.4 | 29.7 | 35.3 |
| NFC | 24.3 | 32.7 | 38.8 | 35.4 |
| Fat | 1.30 | 1.13 | 3.03 | 2.43 |
| Starch | 5.8 | 13.4 | 21.4 | 17.8 |
| Ca | 0.86 | 0.96 | 0.92 | 0.95 |
| P | 0.37 | 0.36 | 0.36 | 0.39 |
| Mg | 0.42 | 0.38 | 0.20 | 0.22 |
| K | 1.42 | 1.25 | 1.31 | 1.48 |
| Co (mg/kg) | 1.65 | 1.71 | 1.88 | 1.69 |

[a]ADF, acid detergent fiber; CP, crude protein; DM, dry matter; NDF, neutral detergent fiber; NFC, nonfiber carbohydrate.

[b]The cows were divided into four groups as follows: (i) dry group (cows between 51 and 33 days before the date of calving), (ii) close-up group (cows between 12 and 0 days before the date of calving); (iii) group 1 (lactating cows; days in milk averaging 125); and (iv) group 2 (lactating cows; days in milk averaging 320).

[c]On a dry matter (DM) basis, the minerals contained per kilogram in the dry and close-up diets were as follows: 63 g of Ca, 44 g of P, 99 g of NaCl, 162 g of Mg, −350 mEq of dietary cation-anion difference (DCAD), 1,210 mg of Cu, 3,307 mg of Mn, 4,463 mg of Zn, 49 mg of Se, 38 mg of Co, 681,430 IU of vitamin A, 184,554 IU of vitamin D, and 12,219 IU of vitamin E.

[d]The minerals contained per kilogram for the lactation diets fed to groups 1 and 2 were as follows: 93 g of Ca, 49 g of P, 111 g of Na, 82 g of Cl, 11 g of K, 16 g of S, 55 g of Mg, 524 mg of Cu, 1,660 mg of Mn, 2,968 mg of Zn, 20 mg of Se, 21 mg of Co, 447,811 IU of vitamin A, 56,671 IU of vitamin D, and 2,777 IU of vitamin E.

phyla *Firmicutes*, *Proteobacteria*, and *Bacteroidetes*, although the relative abundance of each phylum varied based on the niche (Fig. 1D). Several OTUs that corresponded to the *Christensenellaceae*, *Lachnospiraceae*, *Ruminococcaceae* families and *Ruminococcus* at the genus level were detected in each of the rumen, feces, and milk niches (Fig. 1E).

**Effect of diet on microbial population composition.** Each animal involved in this study was fed one of four possible different diets based on their stage of lactation (Table 1). Lactating animals were fed either the group 1 ($n = 53$) or group 2 ($n = 21$) diet while dry; animals calving soon were fed either the dry ($n = 11$) or close-up ($n = 7$) diet. The group 1 and group 2 lactation diets were very similar in terms of composition (Table 1). Alpha-diversity, including taxonomic richness (Chao1 $P < 0.001$; Kruskal-Wallis statistic, 34.77) and diversity (Shannon $P < 0.001$; Kruskal-Wallis statistic, 27.35) varied significantly between feed groups at the OTU level. In terms of beta-diversity, NMDS followed by ANOSIM and PERMANOVA analysis of Bray-Curtis dissimilarities based on diet identified significant dissimilarities in the rumen microbiota (ANOSIM, $R = 0.28$, $P < 0.001$; PERMANOVA, $F = 11.78$, $P < 0.001$) (Fig. 2A) but resulted in less dissimilarity in the fecal microbiota (ANOSIM, $R = 0.24$, $P < 0.001$; PERMANOVA, $F = 6.85$, $P < 0.001$) (Fig. 2B). Lactation diets resulted in negligible dissimilarity in the milk microbiome (ANOSIM, $R = 0.09$, $P < 0.07$; PERMANOVA, $F = 1.54$, $P < 0.054$); however, this finding is based on a comparison of only group 1 and group 2 diets since

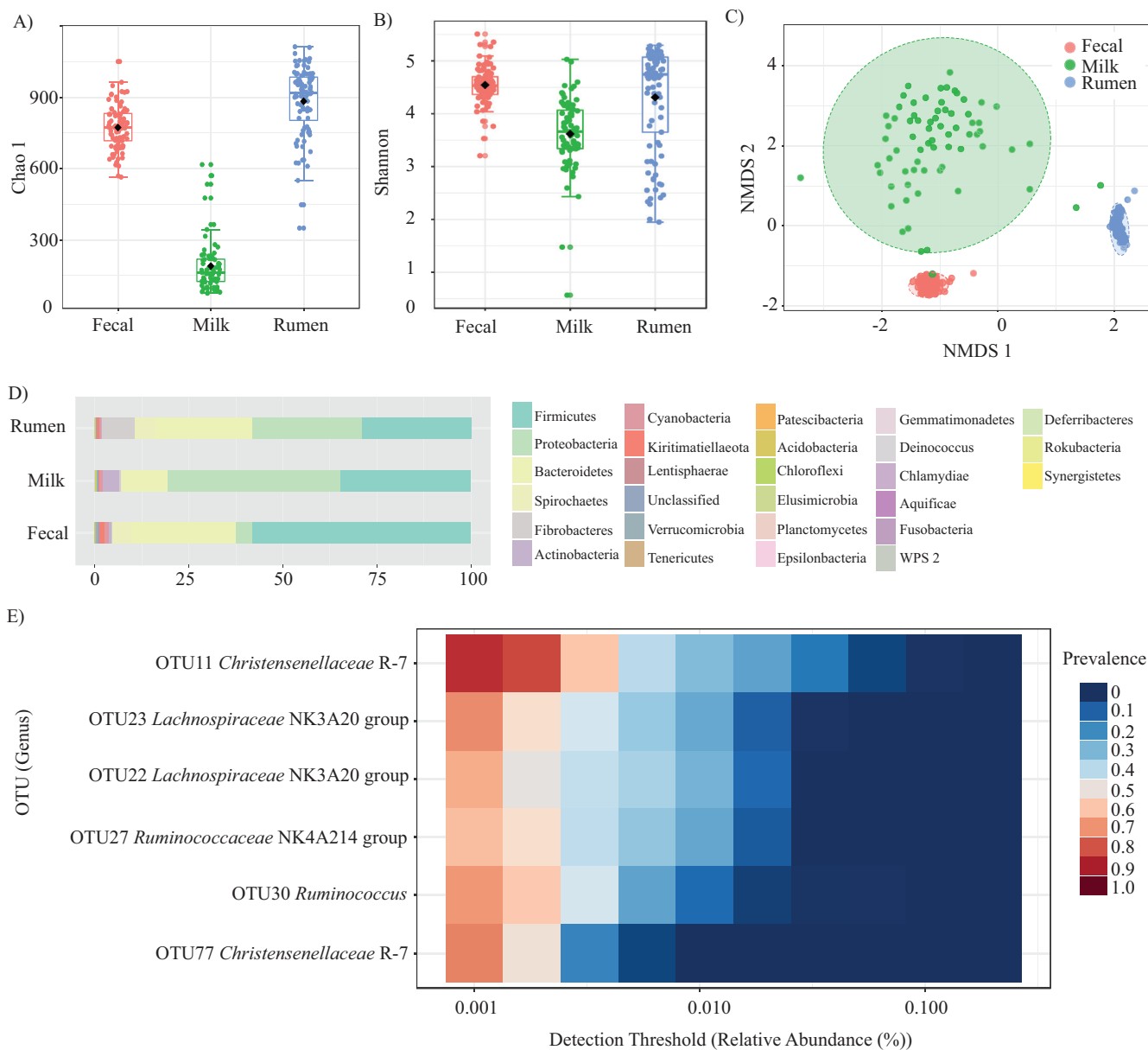

**FIG 1** Bovine microbiota. (A) The alpha-diversity metrics indicate that the species richness is higher in the rumen and fecal samples than in raw milk based on the Chao1 index. (B) Shannon's index of diversity was also calculated and indicated that overall diversity varied between the niches. (C) NMDS ordination illustrated the significant dissimilarity that was observed between communities in each of the niches (NMDS stress = 0.06). In addition, the variances observed in the milk microbiota were significantly higher than the variances observed in either the fecal or rumen samples. (D) A stacked bar graph illustrates the relative abundance of each phyla present in rumen, fecal, and milk samples averaged across all samples analyzed for each sample type and indicates that the presence of each of the major taxa (*Firmicutes*, *Proteobacteria*, and *Bacteroidetes*) in each niche. (E) A heat map displaying the detection threshold and prevalence across niches for OTUs that occurred in >60% of all samples (milk, feces, and rumen).

milk samples cannot be collected from dry cows. Most of the dissimilarity observed in the rumen microbiome could be attributed to differences in the microbiota compared between the lactation diet groups 1 and 2 (ANOSIM, $R = 0.22$, $P < 0.001$; PERMANOVA, $F = 18.11$, $P < 0.001$), while no dissimilarity was observed by directly comparing the rumens of animals being fed the dry and close-up diets (ANOSIM, $R = 0.05$, $P < 0.255$; PERMANOVA, $F = 1.71$, $P < 0.022$). The PERMANOVA value was actually larger when comparing group 1 and 2 diets than when comparing all four diets. This observation is likely due to the low dispersion observed from animals fed the group 2 diet, as this PERMANOVA is sensitive to low dispersion (27). Linear discriminant analysis (LDA) of effect size (LEfSe) identified that the phyla that were present at different abun-

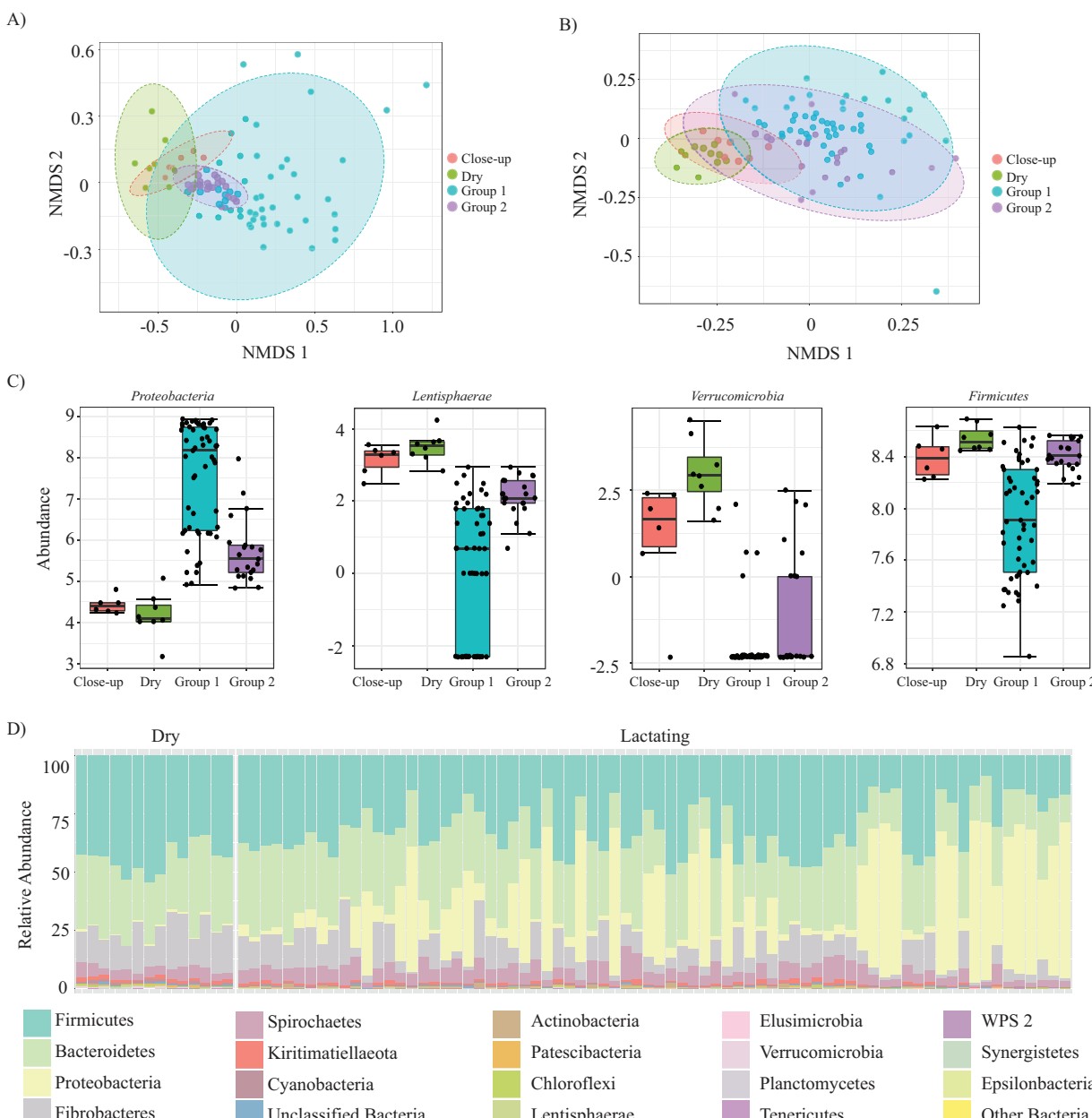

**FIG 2** Variance in the bovine microbiome based on diet and physiological stages. (A and B) NMDS ordination is used to illustrate dissimilarity of rumen samples based on the diet fed to each animal (NMDS stress = 0.09) (A). The rumen showed a dissimilarity based on the diet that was greater than the dissimilarity that was observed in fecal samples (NMDS stress = 0.16) based on diet (B), or milk samples where no dissimilarity was observed (not shown). (C) Log-transformed counts of the statistically significant phyla that were responsible for the dissimilarity based on feed group were *Proteobacteria*, *Lentisphaerae*, *Verrucomicrobia*, and *Firmicutes*. (D) A stacked bar plot illustrates that lactating animals had a significantly higher abundance of *Proteobacteria* in their rumens than their dry counterparts.

dances in the rumen based on diet were *Proteobacteria* ($P < 0.001$, false-discovery rate [FDR] = $8.67 \times 10^{-10}$, LDA score = 3.23), *Lentisphaerae* ($P < 0.001$, FDR = $8.73 \times 10^{-10}$, LDA score = 1.23), *Verrucomicrobia* ($P < 0.001$, FDR = $8.73 \times 10^{-10}$, LDA score = 1.21), and *Firmicutes* ($P < 0.001$, FDR = $1.90 \times 10^{-7}$, LDA score = 3.03) (Fig. 2C).

The rumen and fecal microbiota were also compared based on lactation status, which essentially combines the dry and close-up diet groups into the dry category and the group 1 and 2 diets into the lactating category. The dissimilarity in the rumen (ANOSIM, $R = 0.27$, $P < 0.001$; PERMANOVA, $F = 12.863$, $P < 0.001$) based on lactation was similar to that observed when all feed ration groups were compared. The dissim-

**TABLE 2** Estimated R correlation matrix for vitamin B$_{12}$ measurements

| Factor | Pearson correlation coefficient ($n = 72$) comparing factors[a] | | | | |
| --- | --- | --- | --- | --- | --- |
| | Rumen | Feces | Plasma | Milk concn | Milk yield |
| Rumen | 1.00000 | 0.23648 | 0.22726 | −0.07291 | 0.24076 |
| | | 0.0455 | 0.0549 | 0.5427 | 0.0416 |
| Feces | 0.23648 | 1.00000 | 0.17602 | −0.07786 | 0.07463 |
| | 0.0455 | | 0.1391 | 0.5156 | 0.5332 |
| Plasma | 0.22726 | 0.17602 | 1.00000 | 0.24297 | 0.34962 |
| | 0.0549 | 0.1391 | | 0.0397 | 0.0026 |
| Milk concn | −0.07291 | −0.07786 | 0.24297 | 1.00000 | 0.42641 |
| | 0.5427 | 0.5156 | 0.0397 | | 0.0002 |
| Milk yield[b] | 0.24076 | 0.07463 | 0.34962 | 0.42641 | 1.00000 |
| | 0.0416 | 0.5332 | 0.0026 | 0.0002 | |

[a]Probability $> |r|$ under the null hypothesis (H0): rho = 0. The top number indicates correlation ($r$), while the bottom number is probability ($P$).
[b]Milk yield refers to the concentration of vitamin B$_{12}$ measured in milk, multiplied by the volume of milk produced by the animal at the morning milking on the day when the sample was taken.

ilarity of fecal samples (ANOSIM, $R = 0.49$, $P < 0.001$; PERMANOVA, $F = 14.97$, $P < 0.001$) was higher when samples were compared based on lactation status than when they were compared based on diet. LEfSe was used to determine that the dissimilarity identified in the rumen resulted from a higher abundance of *Proteobacteria* ($P < 0.001$, FDR $= 4.72 \times 10^{-8}$, LDA score $= 3.1$) in lactating animals and a higher abundance of *Lentisphaerae* ($P < 0.001$, FDR $= 4.72 \times 10^{-8}$, LDA score $= -1.14$) and *Verrucomicrobia* ($P < 0.001$, FDR $= 2.09 \times 10^{-9}$, LDA score $= -1.03$) in dry animals (Fig. 2C). Fecal samples from lactating animals also had more *Proteobacteria* ($P < 0.001$, FDR $= 1.40 \times 10^{-8}$, LDA score $= 2.47$), while dry animals had a higher abundance of *Lentisphaerae* ($P < 0.001$, FDR $= 1.33 \times 10^{-8}$, LDA score $= -1.98$), *Verrucomicrobia* ($P < 0.001$, FDR $= 6.42 \times 10^{-8}$, LDA score $= -1.87$), and *Tenericutes* ($P < 0.001$, FDR $= 3.90 \times 10^{-8}$, LDA score $= -1.69$) (Fig. 2D). Based on the level of dissimilarity between animals fed different diets, all rumen and fecal samples taken from dry cows were removed from all analyses aimed at correlating the microbiota composition with vitamin B$_{12}$ concentrations.

**The concentration of vitamin B$_{12}$ in plasma is the best predictor of the abundance of vitamin B$_{12}$ in milk.** Vitamin B$_{12}$ was measured in each rumen sample as well as correlated fecal, plasma, and milk samples (see Table S1 in the supplemental material). The concentration of vitamin B$_{12}$ in the rumen was compared between lactating animals ($n = 74$) and dry animals ($n = 18$), and it was shown that dry animals had a significantly higher concentration of vitamin B$_{12}$ in their rumen than lactating animals ($P = 0.0014$). However, since milk was not available from dry animals for comparison, dry animals were excluded from further comparison and statistical analysis.

A Pearson correlation coefficient test was performed between all vitamin B$_{12}$ concentration results for different sample types to identify correlations between vitamin B$_{12}$ concentrations in different niches (Table 2). To obtain a correlation for each measurement with the total yield of vitamin B$_{12}$ produced by the animal—as opposed to the concentration, for each cow, the concentration of vitamin B$_{12}$ in milk was multiplied by the amount of milk produced during the morning milking. The concentration of vitamin B$_{12}$ in the rumen was not correlated with the concentration of vitamin B$_{12}$ in milk ($r = -0.07$, $P = 0.54$), but was correlated with the total yield of vitamin B$_{12}$ in milk ($r = 0.24$, $P = 0.04$). The concentration of vitamin B$_{12}$ in plasma had the highest correlation with the total yield of vitamin B$_{12}$ in milk of any correlation observed in this investigation ($r = 0.42$, $P = 0.002$) (Table 1).

**Increased abundance of *Prevotella* and *Succinivibrionaceae* correlate with high levels of vitamin B$_{12}$ in the rumen.** To elucidate which bacterial genera are correlated

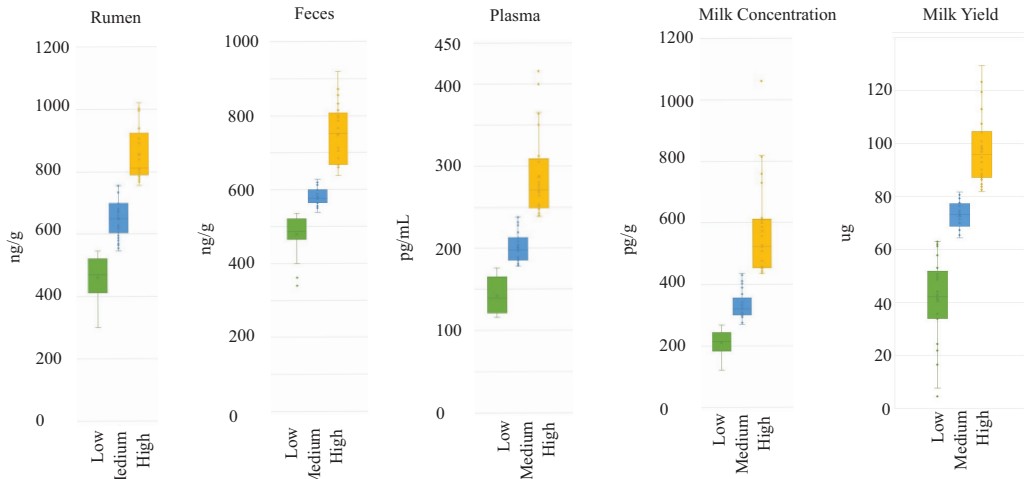

**FIG 3** Vitamin $B_{12}$ concentrations in rumen, fecal, plasma, and milk samples. To facilitate analysis of the microbiota, the samples for each site were categorized as having high, medium, or low concentrations of vitamin $B_{12}$ relative to other samples, so that each category contained approximately 33% of the samples. The total milk category is the yield of vitamin $B_{12}$ produced by the animal on the day of sampling, this number was calculated by multiplying the concentration by the volume of milk produced on the day of measurement.

with vitamin $B_{12}$ concentration, each sample was classified as having either high, medium, or low vitamin $B_{12}$ concentrations relative to other samples, where approximately a third of samples were placed into each category (Fig. 3). In the rumen, vitamin $B_{12}$ concentrations ranged from 300.09 to 1,021.39 ng/g. After dividing samples into three approximately evenly sized groups, rumen samples with a concentration of >735.89 ng/g vitamin $B_{12}$ were classified as having a high concentration ($n = 22$), samples with <563.83 ng/g were classified as having low vitamin $B_{12}$ ($n = 25$), and rumen samples with a concentration that fell between 563.83 and 735.89 ng/g were excluded from this analysis (Table S2). It was reasoned that analyzing the samples with ruminal concentrations of vitamin $B_{12}$ at either end of the continuum would provide the best opportunity to identify which bacterial genera are well correlated with this parameter. The 48 samples included in this analysis were rarified to 17,274 sequences, and Good's coverage at this sequencing depth ranged from 98.5 to 99.5. Both community richness (Chao1 $P < 0.001$; Kruskal-Wallis statistic, 117) and diversity (Shannon $P < 0.001$; Kruskal-Wallis statistic, 104) were higher, at the OTU level, in the rumen samples of animals that had a low concentration of vitamin $B_{12}$ (Fig. 4A and B). Generalized linear models were used to examine diet as a covariate to vitamin $B_{12}$ concentration, where the Shannon index was considered the normally distributed metric. Vitamin $B_{12}$ concentration had a slightly negative effect of Shannon diversity ($P = 0.0334$). The level of dissimilarity between communities based on vitamin $B_{12}$ concentration was low (ANOSIM, $R = 0.29$, $P < 0.001$; PERMANOVA, $F = 8.99$, $P < 0.001$) (Fig. 4C). When diet was considered a covariate to vitamin $B_{12}$ abundance in the rumen using a PERMANOVA analysis of the Bray-Curtis distance using the R Vegan package, and beta-diversity did not vary significantly between high-vitamin $B_{12}$ and low-vitamin $B_{12}$ samples. Only 14 to 17% of variance could be explained by vitamin $B_{12}$ or diet (Table S3). LEfSe was used to identify which genera were significantly differentially abundant based on vitamin $B_{12}$ concentration. *Prevotella* was correlated with high concentrations of vitamin $B_{12}$ in the rumen ($P < 0.001$, FDR $= 2.65 \times 10^{-5}$), while the abundance of a single *Bacteroidetes* OTU that could not be identified beyond the phylum level (*Ruminiclostridium*, *Butyrivibrio*, *Succinivibrionaceae*, and *Succinimonas*) were each correlated with lower concentrations of vitamin $B_{12}$ ($P < 0.001$, FDR $< 0.01$) (Fig. 4D and Table S4) (28). LEfSe was also used to rank correlated genera by effect size, and based on this analysis, *Succinivibrionaceae* and *Prevotella* were correlated with high vitamin $B_{12}$, while *Lachnospiraceae*, *Christensenella*, *Prevotella*, and *Fibrobacter* were each correlated with vitamin $B_{12}$ (Fig. 4D).

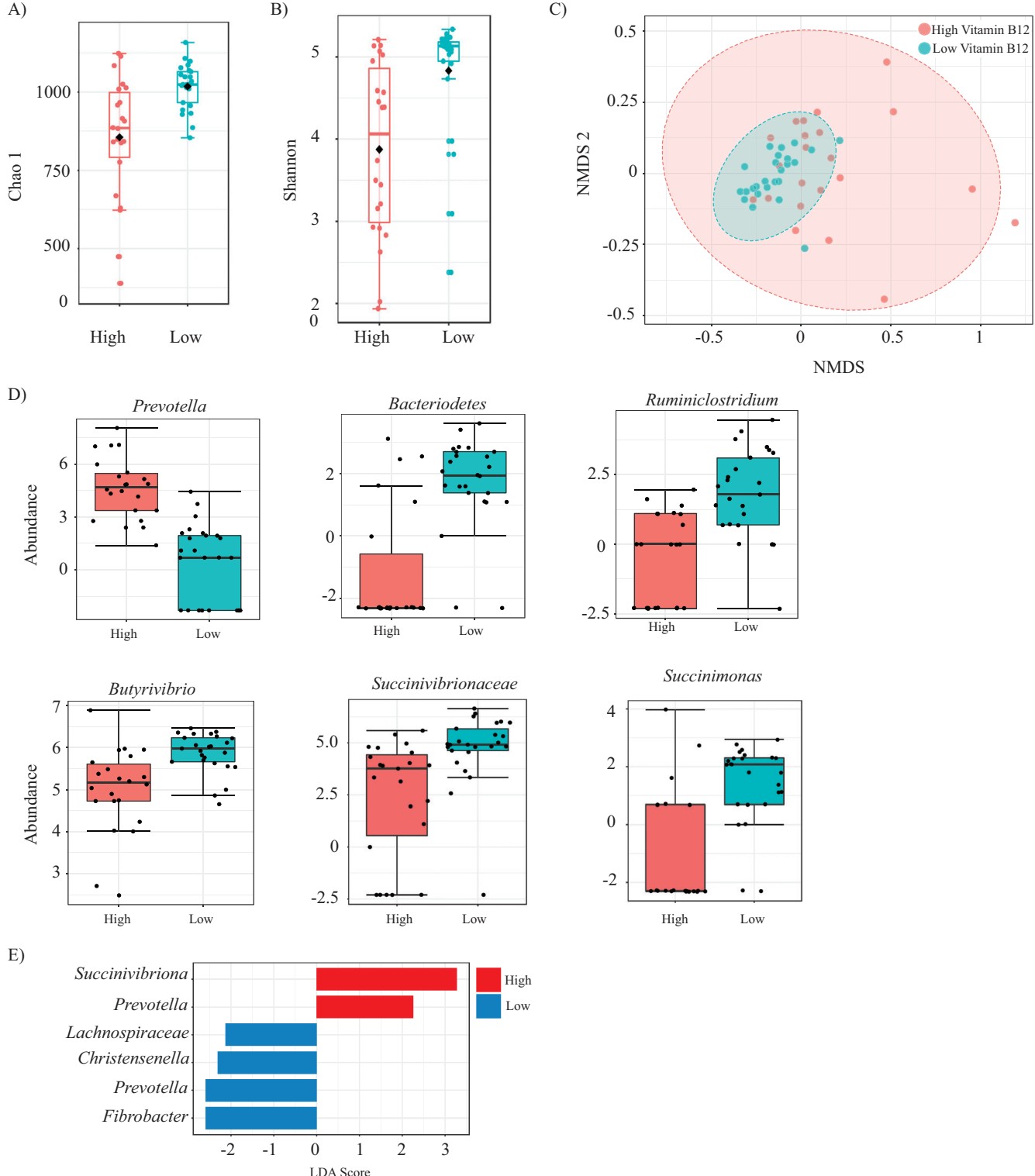

**FIG 4** The rumen microbiota correlated with ruminal concentration of vitamin B$_{12}$. (A and B) The alpha-diversity in the bovine rumen, as measured by both richness (A) and diversity (B) is significantly different between animals when comparing animals that had either a high or low concentration of vitamin B$_{12}$ in the rumen. (C) However, the beta-diversity is not significantly dissimilar between communities based on a comparison of vitamin B$_{12}$ concentrations. (D) At the genus level, the abundance of *Prevotella* is significantly higher in rumen samples with a high concentration of vitamin B$_{12}$, while *Bacteriodetes*, *Ruminiclostridium*, *Butyrivibrio*, *Succinivibrionaceae*, and *Succinimonas* were each correlated with low ruminal concentrations of vitamin B$_{12}$. (E) In some instances, different phyla were correlated with vitamin B$_{12}$ abundance when these features were selected based on effect size, rather than *P* value. *Prevotella* was again correlated with increased levels of vitamin B$_{12}$, while a different OTU classified at the genus level *Prevotella* was correlated with decreased levels of vitamin B$_{12}$.

A LASSO regression analysis was performed using vitamin $B_{12}$ abundance while controlling for diet, where the data were randomly split with 50% of data binned into a training model and 50% of the data put into a testing model. The accuracy of the final model was 0.6521. Using a 100-fold cross validation, we evaluated the frequency of how often genera were selected. The genera that were correlated with vitamin $B_{12}$ abundance while controlling for diet were *Lachnospiraceae* (coefficient value, −0.0066), *Ruminococcus* (coefficient value, −0.013), and *Succinivibrionaceae* (coefficient value, −0.0002).

**Specific taxa were not correlated with vitamin $B_{12}$ abundance in feces or milk.** To identify bacterial genera that were correlated with vitamin $B_{12}$ concentration in fecal samples, we divided samples into three groups based on the concentration of vitamin $B_{12}$ (Fig. 3). In fecal samples, vitamin $B_{12}$ concentrations ranged from 340.16 to 920.31 ng/g. After dividing samples into three approximately evenly sized groups, fecal samples with a concentration of >637.37 ng/g vitamin $B_{12}$ were classified as having a high concentration ($n = 25$), samples with <536.50 ng/g were classified as having low vitamin $B_{12}$ ($n = 25$), and fecal samples with a concentration that fell between 536.50 and 637.37 ng/g were excluded from this analysis (Table S5). The 50 samples included in this analysis were rarified to 16,376 sequences per sample, and Good's coverage was >98.9 for each sample. Neither community richness (Chao1 $P = 0.09$; Kruskal-Wallis statistic, 368) nor diversity (Shannon $P = 0.367$; Kruskal-Wallis statistic, 332) differed based on vitamin $B_{12}$ concentration at the OTU level. In terms of beta-diversity, the level of dissimilarity in the fecal microbiota based on vitamin $B_{12}$ concentration was low (ANOSIM, $R = 0.009$, $P < 0.283$; PERMANOVA, $F = 1.26$, $P < 0.169$), and LEfSe analysis was unable to identify any specific taxa that were correlated with the vitamin $B_{12}$ concentration found in the feces (Table S6).

In mammals, vitamin $B_{12}$ from plasma is partially absorbed by the mammary gland and secreted in milk. We attempted to determine whether a specific bacterial taxon residing in the mammary gland could be correlated with the ultimate abundance of vitamin $B_{12}$ in this niche. For each cow, the concentration of vitamin $B_{12}$ in milk was multiplied by the amount of milk produced during the morning milking. Samples where cows had a high concentration of vitamin $B_{12}$ in plasma but had a small amount secreted in the milk ($n = 24$) were compared to samples from cows that had a high concentration of vitamin $B_{12}$ in plasma and a large amount of vitamin $B_{12}$ secreted in milk ($n = 23$) (Table S7). The 47 samples included in this analysis were rarified to 11,052 sequences per sample, and Good's coverage was >99.0 for each sample at this sequencing depth. In the milk samples, neither community richness (Chao1 $P = 0.93$; Kruskal-Wallis statistic, 269) nor diversity (Shannon $P = 0.63$; Kruskal-Wallis statistic, 242) differed based on vitamin $B_{12}$ concentration at the OTU level. In terms of beta-diversity, the milk microbiota displayed no dissimilarity based on vitamin $B_{12}$ concentration (ANOSIM, $R = 0.02$, $P < 0.191$; PERMANOVA, $F = 1.15$, $P < 0.23$), and LEfSe analysis was unable to identify any specific taxa that were correlated with vitamin $B_{12}$ found in milk (Table S8).

## DISCUSSION

Elucidating and understanding correlations between the bovine microbiota and nutritional qualities of dairy products are a first step toward developing strategies to manipulate the microbiota to provide a naturally stable source of vitamin $B_{12}$ and enhance the public's perception regarding milk consumption as an ideal source of this vitamin. In this study, we attempted to identify individual members of the bacterial population present in the bovine rumen, feces, and raw milk that could be correlated with the level of this vitamin. Our results indicated that the overall abundance of vitamin $B_{12}$ in the rumen is better linked to the absence of high vitamin $B_{12}$ consumers, such as *Bacteroidetes*, than to the presence of efficient vitamin $B_{12}$ producers—which were not clearly identified.

Both feed composition and lactation stage are responsible for determining the composition of the rumen microbiota, although as we have pointed out, these two

factors are confounded by each other (29–31). In our investigation, *Proteobacteria* was observed to be more abundant in lactating cows than in dry cows. Other studies have observed increased *Proteobacteria* in the rumens of animals being fed corn silage and alfalfa silage (32). All of the diets in our study included corn silage at various amounts (Table 1). Only the lactation diets in our study included a legume-grass silage that contained both alfalfa and timothy grass silage, and therefore, this component may have contributed to the expansion of *Proteobacteria*.

The concentration of vitamin B$_{12}$ in milk is known to vary widely among both herds and individual animals based on a variety of factors. It is known that genotype (33), days in milk (DIM) (7, 20), and diet (7, 33, 34) each influence the level of vitamin B$_{12}$ in milk. We observed a range of vitamin B$_{12}$ concentrations in both plasma and milk that were slightly more narrow than those observed previously in the combined literature on Holstein cows (35). The narrow range of milk vitamin B$_{12}$ concentrations observed in our study is due at least in part to the fact that we did not sample colostrum, which tends to have a sevenfold-higher vitamin B$_{12}$ content compared to milk, and because our animals were part of the same herd, they each had access to the same rations and were controlled by the same management practices (33). The concentration of vitamin B$_{12}$ in milk was positively correlated using Pearson correlation to the concentration of vitamin B$_{12}$ in plasma ($P = 0.34$), although this correlation was slightly lower than was observed in a previous study ($P < 0.42$) (29).

In lactating Holstein cows with a high ruminal concentration of vitamin B$_{12}$, based on statistical significance calculated by LEfSe, the genus *Prevotella* was found at an increased abundance, while the phylum *Bacteroidetes*, the family *Succinivibrionaceae*, and the genera *Ruminiclostridium*, *Butyrivibrio*, and *Succinimonas* were each identified at higher abundances in animals with low vitamin B$_{12}$ concentrations. When the genera correlated with vitamin B$_{12}$ in the rumen were ranked based on the effect size, different OTUs corresponding to *Succinivibrionaceae* and *Prevotella* were found to be correlated with increased vitamin B$_{12}$ concentration. Each of these taxa has previously been identified as a member of the rumen microbiome active in digestion (29). In the rumen, *Prevotella* displays a nutritional versatility where several different sugars, amino acids, and small peptides can be used to support growth (36), *Ruminococcus* breaks down fibrous plant materials to produce acetate, formate, and succinate (37), and *Butyrivibrio* can break down both fiber and protein (38). Members of the *Prevotella* genus are not known to produce vitamin B$_{12}$, and vitamin B$_{12}$ supplementation is required in *Prevotella*-specific growth media (39, 40). Therefore, it is likely that *Prevotella* is not producing the vitamin in the rumen but instead is taking advantage of high ruminal concentrations of vitamin B$_{12}$ for increased growth and proliferation. *Prevotella* has also been found in higher abundance in the rumens of dairy cows producing high levels of vitamin B$_{12}$ (32). A member of the *Bacteroidetes* phylum that could not be identified at a finer taxonomic resolution (*Prevotella* is also a member of the *Bacteroidetes* phyla) was correlated with a low concentration of vitamin B$_{12}$ in the rumen. This phylum has been demonstrated to be excellent at acquiring vitamin B$_{12}$ from the environment and can rapidly outcompete bacteria that are less efficient at acquiring vitamin B$_{12}$ under vitamin B$_{12}$-limited conditions (41). Some members of the *Bacteroidetes* phylum can readily bind vitamin B$_{12}$ by way of a surface-exposed lipoprotein at femtomolar efficiency, thus effectively even removing it from bound IF (41). Based on the observations in the current study, the direction of the correlation is unknown—it is unclear whether the concentration of vitamin B$_{12}$ in the rumen is helping to drive the composition of the microbiota or whether the composition of the microbiota is affecting vitamin B$_{12}$ concentrations.

Previous work has shown that vitamin B$_{12}$ supplementation can have measurable effects on the composition of the human intestinal microbiome (42), and using corrinoids to manipulate the composition of the microbiome has been previously proposed in the literature (22). However, extensive empirical evidence that demonstrates exactly how corrinoid abundance influences microbial composition, as well as which taxa are affected, is still lacking. Future work should focus on understanding how removal of

apparent key vitamin $B_{12}$ consumers, like *Bacteriodetes*, from a population affects vitamin $B_{12}$ concentration dynamics, since it is possible that other species will metabolically compensate to maintain homeostasis. Little literature exists to aid in elucidating the relationship between *Ruminiclostridium*, *Butyrivibrio*, *Succinivibrionaceae*, and *Succinimonas* and vitamin $B_{12}$, and future work to understand the details of these relationships should also be conducted.

Until very recently, the bovine mammary gland was thought to be sterile (43). However, next-generation sequencing of the 16S rRNA marker gene has revealed that this environment actually hosts a rich bacterial population (43). The origin of the mammary gland microbiota continues to be an area of debate. In humans and cows, the theory of the enteromammary pathway posits that the communities of the mammary gland originated in the intestine and are transported to the mammary glands via the immune system (44). In cows, the enteromammary pathway is favored to explain the presence of rumen and intestinal bacteria in the udders over colonization through environmental contamination, since some strictly anaerobic intestinal bacteria such as *Bifidobacterium* and *Ruminococcus* have been isolated from raw milk (43, 45). The data presented in this paper support the enteromammary pathway hypothesis with additional data beyond what has been previously shown through direct culture work. While direct culture has shown at the genus level that traditionally intestinal bacteria can be isolated from the mammary gland, in the absence of whole-genome sequencing, there can still be questions surrounding the lineage of these isolates and whether they are truly clonal. In the present study, OTUs corresponding to *Christensenellaceae*, *Lachnospiraceae*, *Ruminococcaceae*, and *Ruminococcus* were identified in rumen, feces, and milk samples. Given the OTU resolution of this observation, it is likely these sequences came from the same genetic lineage. In addition, each of these taxa are strictly anaerobic, and therefore, translocation from the intestinal tract to the mammary gland via the environment is unlikely, which supports, but does not definitely prove, the existence of an enteromammary pathway.

**Conclusion.** In this observational study, the concentration of vitamin $B_{12}$ in rumen and plasma were each weakly predictive of the abundance of vitamin $B_{12}$ ultimately found in milk. High concentrations of vitamin $B_{12}$ in the rumen were correlated with the increased presence of *Prevotella*, and lower concentrations of vitamin $B_{12}$ in the rumen were correlated with the increased presence of *Bacteroidetes*, *Ruminiclostridium*, *Butyrivibrio*, *Succinivibrionaceae*, and *Succinimonas*. The ultimate concentration of vitamin $B_{12}$ in the bovine rumen, feces, and milk is controlled by the complex interaction of several factors, including the composition of the microbiota. Future work should focus on defining the exact relationships of each of the bacteria identified in this study with vitamin $B_{12}$ production and destruction.

## MATERIALS AND METHODS

**Sample collection.** Fifty Holstein cows from the dairy herd at the Agriculture and Agri-Food Canada Research Centre (Sherbrooke, Quebec, Canada) and fitted with rumen cannula were involved in this study. All animals were housed in a tie-stall barn, and the cows were fed the diet twice daily for dry cows at 0800 and 1300 h and once daily for lactating cows at 0800 h (Table 2). Milk, fecal, blood, and rumen samples were collected during the same day for each cow, and most animals were sampled twice, with about 3 months separating sampling events. For the purposes of microbiota analysis, each sample was treated as independent. Sampling was performed between May and August 2018. Cows were at different physiological stages of lactation, but each lactating animal was free from mastitis with somatic cell counts (SCC) values lower than 250,000 cells/ml during sampling. There were four different feeding groups associated with physiological stage and milk production of the cow: (i) dry (cows between 51 and 33 days before the actual date of calving), (ii) close-up (cows between 12 and 0 days before the actual date of calving), (iii) group 1 (days in milk [DIM] averaging 125 and morning milk yield averaging 18.2 kg), and (iv) group 2 (DIM averaging 320 and morning milk yield averaging 8.8 kg). Sample collection was performed by trained and qualified members of the Agriculture and Agri-Food Canada Research Centre Team (Sherbrooke, Quebec, Canada).

Milk samples were collected during the morning milking using a sterile technique. Cows were milked twice at 12-h intervals. An initial check of the milk was performed by stripping the teat four or five times, and collecting the milk into a dark-bottomed container to examine the milk for clumps or off-coloring. Then the teat was placed in an iodine predip followed by a 30-s contact time. The teat was then dried with a clean, disposable wipe. Each teat was wiped with 70% ethanol, and the milk was collected

manually using a gloved hand into 50-ml sterile Falcon tubes which were immediately placed on ice. Samples were stored at –20°C until analysis. Another milk sample was taken from calibrated in-line milk meters for vitamin B$_{12}$ concentration analysis.

Whole-rumen digesta samples were collected before the morning meal by the method of Rico et al. (50). Briefly, digesta samples were collected sequentially from five different sections of the rumen: (i) cranial dorsal, (ii) cranial ventral, (iii) central, (iv) caudal dorsal, and (v) caudal ventral, through a ruminal cannula, approximately 250 ml per section, and mixed by gloved hand. The composite sample (approximately 250 ml) was lyophilized (Virtis, SP Scientific, Warminster, PA) and ground in a Wiley mill grinder (A. H. Thomas Co., Philadelphia, PA) through a 1-mm sieve and then placed in long-term storage at –20°C.

Feces were collected by direct rectum grab sampling. The rectal wall was massaged to stimulate rectal evacuation, and the resulting feces were collected. Samples were lyophilized (Virtis, SP Scientific) and ground in a Wiley mill grinder (A. H. Thomas Co.) through a 1-mm sieve. Samples were stored at –20°C until processing.

Blood samples were taken by caudal venipuncture using a Vacutainer system (Becton, Dickinson and Co., Franklin Lakes, NJ). Blood samples were collected in tubes with the anticoagulant EDTA for vitamin B$_{12}$ analyses. Blood samples were centrifuged within 1 h of collection for 15 min at 3,000 $\times$ $g$ and 4°C. Samples were stored at –20°C until analysis.

**Vitamin B$_{12}$ quantification.** An extractive solution was prepared by dissolving 13 g of disodium hydrogen phosphate (Fisher Scientific, Ottawa, Ontario, Canada), 12 g of citric acid (Sigma-Aldrich, Oakville, Ontario, Canada), and 10 g of sodium metabisulfite (Fisher Scientific) in 1 liter of ultrapure water. For fecal and rumen samples, 0.1 g of solid material was suspended in 20 ml of extractive solution, and 150 $\mu$l of 1.0 M sodium cyanide (Sigma-Aldrich) was added before autoclaving at 100°C for 10 min (35). The tubes were then cooled at room temperature, and the pH was adjusted between 6.2 to 6.5 with 3.3 M hydrochloric acid. The volume of the solution was made up to 30 ml by the addition of ultrapure water, and the samples were centrifuged (3,000 $\times$ $g$, 10 min, 4°C) to remove any additional solids. A volume of 200 $\mu$l of the supernatant was used to determine the concentration of the biologically active form of vitamin B$_{12}$ using the SimulTRAC-SNB Vitamin B12/Folate RIA kit for the quantitative determination (SimulTRAC-S Vitamin B12 [Co57]/Folate [I125]; MP Biomedicals, Solon, OH). Briefly, a SimulTRAC-SNB Binder that contains purified porcine intrinsic factor and radioactive vitamin B$_{12}$ (cyanocobalamin) acts as a tracer to compete against vitamin B$_{12}$ in the sample. All samples were analyzed in duplicate, and the values were averaged. The interassay coefficients of variation were 3.8% for rumen and 3.3% for feces. Milk samples were prepared according to the protocol described by Duplessis et al. (35). Milk and plasma samples were also analyzed in duplicate by radioassay using the SimulTRAC-SNB Vitamin B12/Folate RIA kit (MP Biomedicals), and the results were averaged. The interassay coefficients of variation were 3.7% for milk and 4.0% for plasma.

**Bacterial DNA extraction and isolation.** A total of 200 mg of freeze-dried rumen and fecal sample were used for DNA extraction using the Sox Soil DNA Extraction kit (Metagenom Inc. Waterloo, Ontario, Canada) according to the manufacturer's protocol, which includes a bead beating step. Purified DNA was stored at –20°C.

Milk samples were thawed overnight at 4°C and then gently homogenized using a vortex mixer (Scientific Industries). Three milliliters of milk from each quarter was combined in a 15-ml sterile conical tube to obtain a composite sample from each cow to analyze. Six milliliters of the composite milk sample was centrifuged at 17,900 $\times$ $g$ for 5 min to pellet bacterial cells and remove fat from the milk sample. The pellet was used for bacterial DNA extraction via the Sox Soil DNA Extraction kit (Metagenom Inc.) using the standard protocol (200 mg) described by the manufacturer. Purified DNA was stored at –20°C.

**16S rRNA-targeted amplicon sequencing.** Illumina MiSeq paired-end sequencing was used to determine the bacterial community composition of each sample using the V4 region of the 16S rRNA gene as a proxy. The V4 region of the 16S rRNA gene was amplified from the mixed microbial DNA sample using a set of custom primers (F548 and R806) described by Kozich et al. (46). The set of custom primers consisted of the same forward and reverse primers labeled with different barcodes, such that there were 8 different forward primers and 12 different reverse primers. Each sample was amplified via a unique combination of forward and reverse primers so that the sequences could be demultiplexed after sequencing (46). Each sample was amplified via PCR using the HotStar *Taq* (Qiagen) with 25 cycles of PCR, with 1 cycle consisting of 95°C for 20 s, 55°C for 15 s, and 72°C for 1 min, followed by a final extension at 72°C for 7 min and then holding the samples at 4°C. The PCR products were purified using AMPure XP beads (Beckman Coulter) according to the manufacturer's instructions and quantified using the Quant-iT dsDNA high-sensitivity assay kit (ThermoFisher). Any amplicons that were found to have a concentration of less than 1.5 ng/$\mu$l were reamplified. After quantification, the amplicons for each individual sample were independently diluted using DNA/RNA-free water to a final concentration of 1.5 ng/$\mu$l. A sample library was prepared by combining 3 $\mu$l of each sample. The 16S rRNA amplicons were sequenced using the MiSeq and a 500-cycle V2 reagent kit (Illumina).

**Statistical analysis.** Proc CORR of SAS was used to calculate a Pearson correlation coefficient to estimate the correlation between rumen-milk, feces-milk, plasma-milk, rumen-plasma, and rumen-feces vitamin B$_{12}$ concentrations. The following parameters were considered: effect size among samples ($r \geq$ 0.5 for large effect size, $r \geq 0.3$ for medium effect size, $r \leq 0.1$ for low effect size, $r = 0$ for no correlation) and a significant $P$ value of $<0.05$. Rumen and fecal samples with no corresponding milk sample (due to dry lactation periods) were excluded from the correlation analysis.

**Analysis of 16S rRNA targeted amplicon.** Sequence data were processed using mothur version 1.39.5 (46). A full and detailed description of each command performed as well as parameters used can be found in Text S1 in the supplemental material (47). OTU selection was performed using the SILVA

v.132 reference database. A similarity of 97% was used as a cutoff to perform the cluster process. Sequences from each sample type were analyzed together, and sequences that were not observed at least 20 times were not included in the analysis. Alpha-diversity and beta-diversity analysis were calculated using Marker Data Profiling (MDP) analysis on the MicrobiomeAnalyst web platform (28, 47). This analysis requires consensus taxonomy and the associated shared file, which are each direct outputs of mothur. In each analysis, the samples included were rarified to the sample with the lowest number of sequences.

Significant differences between samples in alpha-diversity were assessed using Kruskal-Wallis within the MicrobiomeAnalyst platform. Beta-diversity was compared between samples using both permutational multivariate analysis of variance (PERMANOVA) and analysis of similarity (ANOSIM) performed on the MicrobiomeAnalyst platform. To identify specific OTUs that were differentially abundant in samples with high and low concentrations of vitamin $B_{12}$, linear discriminant analysis effect size (LEfSe) was performed separately for each comparison through MicrobiomeAnalyst (28, 47). LEfSe employs a Kruskal-Wallis rank sum test to detect features with significant differential abundance genus with regard to binary groups, followed by a linear discriminant analysis (LDA) to evaluate the effect size of differential abundance OTUs. Correlations were reported as significant if they met the following criteria: adjusted $P$ value cutoff of 0.05, log LDA score of 2.0, and an FDR of 0.05.

**Ethics approval.** The experimental protocol was approved by the Institutional Committee for Animal Care of the Sherbrooke Research Centre according to the guidelines of the Canadian Council on Animal Care (48). Care and use of cows followed the recommended code of practice of the National Farm Animal Care Council (49).

**Data availability.** The sequencing data sets generated and analyzed during the current study are available from the Sequence Read Archive (SRA) under BioProject accession number PRJNA527029.

## SUPPLEMENTAL MATERIAL

Supplemental material is available online only.

**TEXT S1**, DOCX file, 0.1 MB.
**TABLE S1**, DOCX file, 0.02 MB.
**TABLE S2**, DOCX file, 0.01 MB.
**TABLE S3**, DOCX file, 0.01 MB.
**TABLE S4**, DOCX file, 0.02 MB.
**TABLE S5**, DOCX file, 0.01 MB.
**TABLE S6**, DOCX file, 0.02 MB.
**TABLE S7**, DOCX file, 0.01 MB.
**TABLE S8**, DOCX file, 0.02 MB.

## ACKNOWLEDGMENTS

We are grateful to Jasmin Brochu and Liette Veilleux (Agriculture et Agroalimentaire Canada, Sherbrooke, Quebec, Canada) for their technical help. We acknowledge Marie-Ève Bouchard, Émélie Bell, Matthew Suitor, Étienne Viens, and the barn staff (Agriculture et Agroalimentaire Canada, Sherbrooke, Quebec, Canada) for their help with animal collection and care of cows throughout the experiment.

This project was funded by an Op+Lait Subventions *Nouvelles Initiatives* grant to J.R., R.G., M.D., and C.L.G. J.F.-L. received stipend support from the NSERC CREATE in Milk Quality.

J.F.-L., A.B., and C.R. extracted and sequenced bacterial DNA samples and analyzed the sequences. M.D., C.L.G., L.B., and W.P. collected samples. D.E.R., M.D., C.L.G., R.G., L.B., and W.P. designed the sampling plan and technique. R.I.C. provided statistical analysis of the relationships between the vitamin $B_{12}$ concentrations at each body site. C.L.G., M.D., R.G., and J.R. conceived the project idea and wrote the manuscript. All authors read and approved the final manuscript.

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
