## [Reviewer comments · mSystems]

Correlations between the composition of the bovine microbiota and vitamin B12 abundance

Julian Franco-Lopez, Melissa Duplessis, An Bui, Coralie Reymond, William Poisson, Lya Blais, Jasmine Chong, Rachel Gervais, Daniel Rico, Roger Cue, Christiane Girard, and Jennifer Ronholm

Corresponding Author(s): Jennifer Ronholm, McGill University

Review Timeline:

Submission Date:

February 6, 2020

Accepted:

February 6, 2020

Editor: Jack Gilbert

Reviewer(s): The reviewers have opted to remain anonymous.

Transaction Report:

DOI: <https://doi.org/10.1128/mSystems.00107-20>

We would like to thank the reviewers for their time and insightful comments. The manuscript is much better after the suggested revisions.

Reviewer comments:

Reviewer #1 (Comments for the Author):

The manuscript by Franco-Lopez and colleagues describes the variation in the rumen, fecal, and milk microbiota of 50 Holstein cows and how that variation correlates with the concentration of vitamin B12 in rumen, fecal, milk, and plasma samples. The rationale for the study was solid and the researchers posed an interesting question. Namely, given B12 production still seems to vary widely after controlling for diet and genetics, does variation in the animal's microbiome account for variation in B12 concentration in the milk? The data are presented as for an observational study and it would have been interesting to see the same animals tracked over diet changes or multiple lactations to see whether the associations between the microbiome and B12 production were more robust. As it is, the study design and analysis are solid, but there are a number of places where it could be significantly improved without too much effort.

Major comments

1. The manuscript jumps from the Introduction to the Results without any introduction to the study design. Key information is missing. For example, the breed of the cows (Holstein) and their purpose (dairy) is not described until the Methods section at the end of the paper. Although the second paragraph gets into the different diets and management situations in the study, it is not clear how many animals received each diet or were in each management situation. It's also not clear what the multiple samples per cow represent. If there were 50 cows, what are the 92 fecal samples from? If these are biological replicates from a different day or technical replicates then they would need to be included in the statistical models used to test the authors' hypotheses. Overall, I would recommend the opening paragraph of the Results section lay out the experimental design.

The following paragraph has been added to the Results section to lay out the experimental design, as per this reviewer's suggestion:

"In this investigation 92 rumen, blood and fecal samples, and 71 milk samples were collected from 50 Holstein dairy cows fitted with rumen cannula. Sampling was conducted over a 6-month period, and individual animals provided two entire sample sets (rumen, blood, fecal, and milk) at two different time points approximately 3 months apart. Animals that were culled, or experienced an infection requiring antibiotic treatment, were eliminated from the study and may have only contributed 1 sample set. Lactating animals were fed one of two similar lactation diets, group 1 (n=53) or group 2 (n=21) for the duration of the study, and animals that finished lactation during sampling were fed a close-up followed by a dry diet (Table 1). Milk samples are not collected from dry animals, but fecal, blood, and rumen samples were collected from dry animals for the purposes of comparison. The microbiome of each rumen, fecal, and milk sample was characterized via 16S rRNA targeted amplicon sequencing, and the vitamin B12

concentration of each sample was measured. The datasets were combined and bacterial genera that were correlated to vitamin B12 concentration were identified.”

2. The number of reads per sample is a tradeoff with the number of samples. Would the researchers have been able to include more samples and improve the coverage of the three sites for each cow if they had reduced the number of reads? Even 3,000 sequences per sample is an excellent sequencing depth.

We agree that the number of reads *can* be a tradeoff with the number of samples, but it was not in this study. Lactating dairy cows with a fistula are rare, hard to access, and very expensive – we sampled each animal we could access twice to improve sample sizes – but access to fistulated cows was the limiting factor for samples in this study, and therefore, we would not have been able to include more samples by reducing the number of reads.

The required sequencing depth to have adequate Good’s coverage is based on sample type, and in specific, it is based on the amount of species richness in that particular sample type. Other work we are conducting in this area has indicated that 3,000 sequences are indeed more than enough sequencing depth for milk samples. However, even at 20,000 sequences per sample, we routinely only observe a Good’s coverage of 98 for rumen samples. Therefore, 3,000 sequences would not be enough for this more species rich sample type.

3. Much of the analysis is based on analysis of alpha and beta diversity statistics, which are strongly affected by uneven sampling effort. Although the authors indicate they removed samples with fewer than 10,000 sequences (L140), it is not clear whether they rarefied their alpha diversity data for the subsections starting at L138 or L162. This was the approach taken by reference 48 cited in the Methods, but it is not clear that the authors did this. Furthermore, at L506 the authors indicate that the sequence analysis commands and parameters were included in the Supplementary Methods, but this does not seem to accompany the manuscript. The analysis gets confusing later in the Results as the authors state that they rarefied 48 samples to 17,274 sequences for one analysis (L243), 50 samples to 16,376 sequences for another analysis (L265), and 47 samples to 11,052 sequences for another analysis (L281). I would prefer to see the authors pick a single threshold and stay with it across the study. This would be far less confusing and should not change the overall story. Again, a lower threshold may increase the number of samples in each comparisons without impacting sensitivity and possibly increasing their statistical power.

In this study, a total of 263 samples were processed and there were three samples (1 milk and 2 fecal) that did not produce more than 10,000 sequences after having made 2 or in some cases 3 independent DNA extraction and sequencing attempts. These samples each produced significantly less than 10,000 sequences (<500 in one case). DNA amplification was obviously an issue for these samples – although it was unclear why. Therefore, a lower threshold would not have impacted sensitivity or statistical power.

We did rarify our data as in reference 48 for the alpha diversity data. The sentence, “After rarifying to 10,953 sequences, clustering into operational taxonomic units (OTUs) at the 97% similarity threshold resulted in the identification of 5,028 unique OTUs that were observed a minimum of twenty times in at least one sample type” was added to clarify this point.

We did attach our supplementary methods, and it’s not clear to us why this was unavailable to the reviewer for the first revision. The supplementary methods (which includes the sequence analysis commands) have now been included at the end of the manuscript file (before the references) to aid in review.

With regards to picking a single threshold and stay with it across the study. We had considered this; however, since we are using different datasets for each comparison – we did not see a good reason not to use the most sequences possible for each comparison, and it is my understanding that this is the best practice in most scenarios. Otherwise, we are needlessly throwing out good sequence data to simplify the writing in the results section which seems crazy to me. I feel that the writing is clear as to how many sequences were used for each comparison – and what Good’s coverage this resulted in. We tried the same analysis rarifying to 10,000 for each analysis and in one instance the result was different (it was a rumen sample where more sequence depth appears to be important) – so we would prefer to keep the analysis the way that it is as I don’t think that there is anything objectively wrong with the way we did it.

4. In the Introduction, the authors state that variation in B12 production does not seem to be fully driven by genetics or diet (L117); this is part of the rationale for looking at the microbiome in this study. Their analysis deals with diet separately from other factors, rather as a covariate. I would be interested in seeing the authors use a statistical model that includes diet and some element of genetics as covariates. This is possible with PERMANOVA for comparison of beta diversity measures (adonis in the R vegan package can do this) and with GLMs for alpha diversity metrics that have been transformed to be normally distributed.

This is a great suggestion. Unfortunately, beyond the using the Holstein genetic background, we did not collect data on bovine genetics, since this was not the objective of this study. However, as per the suggestion, we did build a statistical model that treats diet as a covariate, and used PERMANOVA for comparison of beta diversity considering diet and vitamin B12 production (using adonis in the R vegan package), and got the following results:

	F-value	R-squared	P-value
PERMANOVA, bray-curtis distance, Vitamin B12 Group	9.5016	0.1712	<0.001
PERMANOVA, bray-curtis distance, Diet	8.0209	0.14848	<0.002
PERMANOVA, bray-curtis distance, Vitamin B12 Group, Diet	10.076	0.1712	<0.001
PERMANOVA, bray-curtis distance, Diet, Vitamin B12 Group	8.7388	0.14848	<0.001

indicating the beta-diversity in the high and low vitamin B12 groups is not very different, even when diet is considered as a covariate.

This analysis has been incorporated into the paper, and the above table has been added to the supplementary information (now Table S3).

The following sentences were added to the manuscript:

Diet (Group 1 or Group 2) was also considered as a covariate to vitamin B12 abundance in the rumen by a PERMANOVA analysis of the Bray-Curtis Distance using adonis in the R Vegan package, and beta-diversity did not vary significantly between high-vitamin B12 and low-vitamin B12 samples. Only 14-17% of variance could be explained by vitamin B12 or diet grouping (Table S3).

GLMs was used to examine alpha-diversity when considering diet and vitamin B12 as covariates. We examined Shannon as the normally distributed metric. We tested to see if the interaction of vitamin B12 and diet impacts alpha diversity using:

glm(formula = shannon ~ VitB_Conc * Diet, data = sampledf)

Coefficients:

	Estimate	Std. Error	t value	Pr(> t)
(Intercept)	5.3145848	0.5626350	9.446	3.76e-12 ***
VitB_Conc	-0.0017049	0.0007764	-2.196	0.0334 *
DietG2	0.1264021	0.9528317	0.133	0.8951
VitB_Conc:DietG2	0.0010470	0.0016033	0.653	0.5171

Signif. codes: 0 '***' 0.001 '**' 0.01 '*' 0.05 '.' 0.1 ' ' 1

AIC: 129.7

Vitamin B12 had a very slightly negative effect on Shannon diversity.

We also did pairwise comparisons between groups and correct for multiple comparisons run Tukey's honest significance test of our ANOVA:

Tukey multiple comparisons of means
95% family-wise confidence level

Fit: aov(formula = shannon ~ VitB_Group + Diet, data = sampledf)

\$VitB_Group

	diff	lwr	upr	p adj
Low-High	0.9310913	0.4323029	1.42988	0.0004883

\$Diet

	diff	lwr	upr	p adj
G2-G1	0.5358069	-0.02491115	1.096525	0.0606131

To describe this analysis, we added the following sentence:

GLMs was used to examine diet (Group 1 or Group 2) as covariates to vitamin B12 abundance, where the Shannon index was considered as the normally distributed metric, and vitamin B12 had a slightly negative effect of Shannon diversity ($p = 0.0334$).

5. The Results spend a lot of space describing correlations between various taxa and B12 concentrations. I think the results would be more robust, if the authors could build a regression model to predict B12 concentrations or classification model to predict their B12 categories based on the relative abundance of the microbiota after controlling for diet. As the analysis stands, it is difficult to parse out what part of the changes in relative abundance are due to B12 or due to diet (or due to genetics).

Also, a great suggestion. We built a regression model of the relative abundances of bacteria to predict High/Low vitamin B12 whilst controlling for diet. We did this using LASSO regression (penalized logistic regression) where less informative variables are forced to 0, and only the most significant variables are kept in the final model. We randomly split the data where 50% went into the training model and 50% went into the testing model. The accuracy of the model was 0.6521. Using a 100-fold cross validation we evaluated the frequency of how often a genus was selected.

The genera that were selected were *Lachnospiraceae* (coef. value -0.0066), *Ruminococcus* (coef. value -0.013), and *Succinivibrionaceae* (coef. value -0.0002). With the exception of *Ruminococcus* the others had been identified as being correlated via LEFSE.

We have added the following text to the manuscript to discuss this:

A LASSO regression analysis was performed using vitamin B12 abundance while controlling for diet, where the data was randomly split with 50% of data binned into a training model and 50% of the data being put into a testing model. The accuracy of the final model was 0.6521. Using a 100-fold cross validation we evaluated the frequency of how often genera were selected. Genera that were correlated to vitamin B12 abundance while controlling for diet were: *Lachnospiraceae* (coef. value -0.0066), *Ruminococcus* (coef. value -0.013), and *Succinivibrionaceae* (coef. value -0.0002).

Other comments

6. It would also help with readability if the Ns were more uniformly listed. For example at L140 they state "a total of 92 bovine fecal samples and corresponding rumen (n=88) and milk samples (n=71)". It would be easier to read if they stated "bovine fecal samples (n=92) and corresponding rumen (n=88) and milk samples (n=71)". Furthermore, in many places it is not clear what the N for each sub-group is. It should not be necessary to comb through the supplementary materials to

find such important information.

We have revised throughout to make the manuscript clearer in regards to what n is, particularly in the results section.

7. In the figures the authors switch between stacked bar charts for average communities and boxplots for individual taxa. This is a bit jarring and I think the authors should stick with boxplots. As an example of why this is important, in Figure 1D, the authors present three stacked bars indicating the average. Yet the "raw" data for the rumen in 2D reveals considerable phylum level variation. The averages in 1D don't seem to mean much and hide considerable information.

We disagree with this comment. Bar charts are an effective way of summarizing an entire community structure, while box plots are very effective to show the variation of a particular taxa between samples. They show different things well. We agree that Figure 1D masks considerable information, but it's information that we later dig into and probe in the other figures. The strength of 1D is that it summarizes the differences between sample types at the Phylum level – which I think has value. Specifically, the value is that even at this high phylogenetic level of phylum significant differences exist between each of our sample types.

8. The y-axes for the boxplots do not make sense. First, the legends describe these as being "Log transformed counts" (e.g. L571). First, these should be relative abundances and not counts as hopefully every sample has the same number of total counts. Second, what power was used for the transformation? I can't believe it was 10, since they have Proteobacteria abundances that are at 9 (Figure 2C) and that would indicate a billion reads. It would be better to put the actual value on the y-axis and space the values in a log scale. It would also be nice for all of the boxplots to have a common range on the y-axis so that it is easier to compare the effect size across taxa.

This is an excellent comment and led to some important additions to the paper – which we feel made it more understandable, but we will have to provide a lengthy explanation of why.

Yes - each sample has the same number of total counts in this figure (they were all rarified to 17,274). Using R, the total counts are considered, which ever counts are 0 are changed to 0.1 for plotting purposes, and then all counts are log transformed. For reference, the code we used to do this was:

```
data <- mbSetObj$analSet$boxdata; # get the filtered counts at the selected taxonomy level
a <- data[,feat]; # get the specific count of the selected feature
ind <- which(a=="0"); # whichever counts are 0 change to 0.1 for plotting
a[ind] <- 0.1;
data$log_feat <- log(a); # log all counts
```

For Prevotella this results in the following transformation of filtered count to log-transformed count:

We could plot the filtered count but plotting the log-transformed count is a much better display of the data. I'm not sure how we would accurately put an actual value on the axis but use the log-transformed spacing, I think this would mis-lead readers.

While we agree that a common range on the y-axis so that it is easier to compare the effect size across taxa would be nice, it's not really a good way of displaying the data – since the absolute (and relative) abundances are so drastically different. As an example, here are the same graphs for Bacteroidetes, rather than the highest abundance being just over 3000, it's just over 30 – it does not make sense to plot these two datasets using the same axis. However, if readers are really interested in these numbers, they are in Table S3 (along with p-values, FDR, averages for the high and low vitamin B12 groups, and the LDA score).

While we were considering this comment, it came to our attention that ranking potential biomarkers based on statistically significant features alone may neglect the importance of features that are differential between the high and low classes based on effect size (which had some differences from using the p-value alone). Therefore, we have included another section in Figure 4 showing the important features based on effect size (also shown below for reviewers' convenience). We have also included a discussion of this data in the Discussion section.

9. The y-axis for the strip charts for alpha diversity metrics in Figures 1 and 4 should include zero.

The y-axis for the strip charts in Figures 1 and 4 now contain zeros.

10. The diameter of the red ellipse in Figure 4C seems inflated. It gives the sense that the low B12 group is a subset of the high B12 group and that seems like a stretch from looking at the data.

The red ellipse in Figure 4C is a 95% confidence ellipse generated by the R software within the MicrobiomeAnalyst tool. It is not artificially inflated. The population variances were indeed greater within the high Vitamin B12 dataset.

11. I don't feel that P and FDR values need to be presented for the same test. The FDR value would be sufficient.

While I agree that the two numbers are redundant, it is my experience that several experts in the field prefer P while several prefer FDR. It takes very little space to display both numbers, so I prefer to leave both included. Other published manuscripts have also done so – so there is precedence.

12. Throughout the manuscript, numbers that are supposed to be superscripts are instead

presented as subscripts.

This has been revised to the best of our abilities – but we only found one instance of this. Based on the comment, I think the reviewer saw more instances of this and it may be an artifact of the submission software?

Reviewer #2 (Comments for the Author):

Franco-Lopez et al. reported the correlation between the composition of different niches of bovine microbiota and vit B12 abundance. The manuscript describes an observational study that was performed in one farm using 50 rumen cannulated dairy cows from a wide range of days in milk. The microbiome study was 16S based, therefore, superficial for the depth of conclusions that authors aimed to reach in this paper (contribution of bacterial species to the vitamin B12 synthesis; as such a large portion of discussion is speculative). Authors described difference among the microbial communities of the rumen, the feces and the milk and come up with strong conclusions that the rumen microbiome or diet or lactation stage do not have any impact on the milk microbiome. In order to come up with such conclusions authors should have performed a controlled study where confounding factors are accounted for.

We did not make the conclusion that the rumen microbiome has an impact on the milk microbiome. But we take the point of this reviewer and have tried to temper down some of our statements throughout the manuscript.

Our data very strongly support that the milk microbiome was not influenced by diet. Although the two diets in question were very (VERY!) similar (Table 1), so this makes a lot sense, and contributes to the overall findings in the study.

We did not comment on the effect of lactation stage on the milk microbiome since you cannot collect milk from a dry cow to study this.

All being said, the intro, methods and result sections were described properly, although the inclusion of dry cows from the beginning of study was questionable. The discussion and especially conclusion section however require revision. I encourage authors to remain focus on their objectives and acknowledge the limitations of their study in order to avoid misdirecting scientific community or the readers who have limited microbiome knowledge. Discussion/conclusions related to the entero-mammary pathway need to be either removed or strong justification be provided. For detailed comments please see below:

We agree that the inclusion of dry cows was questionable – but the data are interesting, and we feel that it should be part of the public domain. The data from dry cows was not included in any of the analyses upon which we drew our conclusions.

We have removed discussion of the entero-mammary pathway in the conclusions. However, since our observations do provide support for this controversial theory, and since we don't think

publishing a separate paper on this observation is appropriate, we would like to leave the report of this confined to a few sentences in the results section and a few sentences in the discussion. We feel this is appropriate. Our data are publicly available, and readers are free to further investigate this aspect if it interests them.

Detailed comments:

Abstract:

L48-71. Abstract lacking basic information about the number of cows that were tested, number of samples that were collected/analyzed, stage of lactation of cows at sampling time point, dietary conditions that were evaluated, method of microbiome evaluation (16S etc, what region). Adding this information will enrich the abstract. Currently 1/3 of abstract has been used for the introduction of the concept. This can be shortened.

Several sentences have been added to the abstract to add the details requested, and several introductory sentences have been removed from the abstract.

L51: ...consumption of dairy products....

The word "of" has been added.

L56:....with milk vitamin B12....

We are not going to add the word milk here, since we also correlated populations to rumen, fecal, and blood concentrations of vitamin B12.

L60:...populations

"s" has been added to "population."

L70: is this supported by your data?

Yes, I think that it is. Firstly, the statement has been tempered with the word "may" to clearly convey that this is speculative. But we have identified several bacterial genera that are correlated to high levels of vitamin B12 in the rumen. Each of these genera appear to consume vitamin B12 based on the literature – and none have been linked to production. Therefore, our working hypothesis for follow up studies being conducted at the moment is that bacterial consumption is more important to variability than production – and that working hypothesis is the result of this study.

Intro:

L74: maybe replace ", and requires" with "in"

"and requires" has been replaced with "in"

L77: cattle instead of dairy cattle (as you mentioned B12 also can be obtained from meat)

The word “dairy” has been removed.

L82: cows and herds

The order of these words has been inverted.

L89: would be good to add where the IF is produced from

The sub-clause “produced by the parietal cells of the stomach” was added to the sentence to describe where the IF was produced.

L103: does this mean that generally speaking the udder of dairy cattle has low efficiency for uptake and secretion of vit B12? So it doesn't matter how much B12 is produced in the rumen if less than 5% of the plasma B12 has been taken and released into milk. Not sure how you are trying to improve the process.

Yes. It means that the udder of the dairy cow has a relatively low efficiency for the uptake and secretion of vitamin B12. However – it does matter how much is produced in the rumen, since although you are losing the vitamin along the cycle, and up-take isn't perfectly efficient, more production ultimately does mean more release in milk.

We are trying to improve the process by understanding it. Collectively, we don't know any of the details surrounding microbial production and consumption in the rumen. We don't know the importance bacterial of consumption in the intestine or udder. This paper is the first from a series (hopefully) from our lab to offer some light on microbial contributions to variation.

P106 up to the end. So this all contradict the fact that animal scientists are trying to improve the efficiency of production or reduce methane emission as increased production of B12 requires greater F:C ratio.

Perhaps. We are providing a brief review of the literature on vitamin B12 production in cows, we have not discussed methane emissions, or milk production efficiency based on volume in this paper and feel that these topics are beyond our scope.

L120. Very few but diverse? Please revise to make the point clear.

What we mean is that a small percentage of species within either domain are capable of producing vitamin B12 (**few**) – but those that are span two domains of life and several phyla within each domain (**but diverse**). We have changed the sentence to the following:

“The microbial synthesis of vitamin B12 is complex, and due to the elaborate structure and high metabolic cost of its synthesis, only a very few, but phylogenetically diverse, bacteria and archaea are known to produce it [22,23].”

L127. What is the benefit of B12 for bacteria itself. Maybe open this up for better understanding of the readers.

We have added several sentences to clarify why bacteria encode vitamin B12:

“Several bacteria and archaea lineages encode enzymes, that perform radical rearrangements or methyl transfer reactions, which require vitamin B12 as a cofactor [23]. Like humans and other eukaryotes, most bacteria require vitamin B12 as a cofactor for methylmalonyl CoA mutase (MCM) and corrinoid-dependent methionine synthase (MetH); the former catalyzes the interconversion of *R*-methylmalonyl-CoA and succinyl-CoA which is a critical step in the metabolism of amino acids, fatty acids, cholesterol, and sugar to propionate [22], while the latter catalyzes the final step in the biosynthesis of methionine [23]. Unlike eukaryotes, some species of bacteria can also use MCM to convert succinyl-CoA to methylmalonyl-CoA a process which ultimately generates CO₂ and has been hypothesized to facilitate respiration in anaerobic environments, like the intestine [24,25]. Vitamin B12 is also required as a co-factor for ethanolamine ammonia lyase which allows bacteria to use ethanolamine, which is commonly found in the gut, as a source of carbon and nitrogen [26].”

L128-130. Not sure exactly what does that mean. If 80% are dependent on exogenous B12 why 25% of them still can make it through de novo synthesis. You mean they don't produce enough through de novo synthesis that still are dependent on exogenous B12.

We mean that 80% use exogenous sources, while only 25% have the genes to make it. This work is based on the number of bacteria that have the genes for production, and the number that have the genes for vitamin B12 up-take. While there is no wet-lab work to directly demonstrate that some don't make enough to support their needs and then need to collect exogenous vitamin B12, these numbers would definitely suggest that. We have tried to clarify the sentence by making the following adjustments:

“Although not known for dairy cows, approximately 80% of the commensals that comprise the human intestinal microbiota requires exogenous vitamin B12 based on the presence of vitamin B12 specific import systems, while less than 25% human commensals can produce the vitamin through de novo synthesis [5].”

L133. But as you mentioned the bottle neck is the uptake of B12 by the mammary and its release into milk. This apparently is more important than B12 production in the gut.

Variance in mammary gland up-take is very important, but it still doesn't explain variance in vitamin B12 concentrations in the rumen, plasma, or if it gets consumed in the mammary gland – which is why we did the study.

Methods:

L402. Table 1, superscripts doesn't match with the footnotes.

The superscripts refer to the footnotes we wish to draw attention to for each column title. Beyond that, we are not sure what this comment is referring to.

L407. SCC wont be a defined term for the audience of this journal. Please spell it out.

SCC is now defined as somatic cell count in the text.

L406-408. So this was an observational study with relatively a small number of cows, from all stage of lactation and on different diets. Hard to conclude anything about diet or lactation stage at the end while there was such a conclusion in the abstract.

We are assuming that this comment is about the following sentence:

“Combined diet and lactation status had significant effects on the composition of the microbiota in the rumen and in the feces.”

Our data very clearly showed different microbiota in the rumen and feces of dry cows. We have enough animals to make this correlation statistically significant. However, we do agree that we have no way of telling if this was because of lactation or diet – since cows are necessarily fed drastically different diets if they are lactating than if they are dry, which is why we worded it the way we did.

L415. Indicate when milk samples were collected? And how many times cows were milk per day.

The following sentence has been added to address this comment:

“Milk samples were collected during the morning milking using a sterile technique. Cows were milked twice again at 12-hour intervals.”

L424. delete 2015.

2015 has been deleted

L457. et al.

A period has been added after *al.*

L464. Indicate whether the DNA extraction kit included a bead-beating step or not.

The sentence has been changed to:

“A total of 200 mg of freeze-dried rumen and fecal sample were used for DNA extraction using the Sox Soil DNA Extraction Kit (Metagenom Inc. Waterloo, ON) according to the manufacturer’s protocol, which includes a bead beating step.”

L466. Creating a composite sample of milk after thawing is a difficult task due to the nature of milk and presence of the fat layer. Samples should have been mixed before frozen.

We kept the samples separate to determine if there were significant differences in the microbiome or vitamin B12 concentrations between the individual udders on a pilot inspection of the samples. This was done because we thought udder microbiota may be consuming vitamin B12. However, there were not significant differences in the microbiome between the individual udders (in healthy cows with normal SCC), so we felt comfortable creating a composite sample. Samples were fully thawed and homogenized via vortexing before mixing we are comfortable with what we did and why.

L469. Min

Minutes has been changed to min.

L472 and other places. Always better to keep DNA samples at -80. The size of ice crystal and the rate of their formation is way higher at -20 and severely impacts the quality of DNA especially when dealing with low load microbial samples such as milk. I think somewhere within the text you should address these limitations, so it doesn't become the norm.

We agree. Everything said in this comment is absolutely correct. However, we have done pilot studies in our lab comparing the 16S rRNA targeted amplicon sequencing results from never frozen milk, and the same sample frozen at -20C for 6 months, 1 year, 2 years, and we will do a 3-year sample as well. After 2 years there are no detectable changes in the 16S rRNA profile (metagenomics may give a different result). Our results are so consistent over the 2-year period, that I think most people in the field will be surprised when we publish the results. Based on my own carefully conducted pilot studies and since these samples were stored for less than 2 years I do not agree that this is a limitation (at least in a milk matrix with 16S rRNA TAS).

L478. Delete 2013

2013 has been deleted.

L491. uL or ul. Just be consistent across the manuscript.

This ul has been changed to uL.

L501. The correct scientific way is that there should be no equal for the upper limit of any p-value. It should be less than 0.05.

This has been corrected throughout the manuscript.

Results

L140. Why such a low cutoff of 10,000 cutoff? You considered this cutoff for all sample types based on milk samples? What limitation this decision can cause down the road?

This cutoff was determined based on Good's coverage for all sample types above 98. For milk and fecal samples, the Good's coverage at 10,000 reads was always above 99, and for rumen samples it was always greater than 98. I don't think this causes any limitations down the road.

L146. Why twenty times? Better give a percentage.

We disagree. The command in Mothur which controls this parameter is based on an absolute number and we set it at 20. Its more accurate to report it this way since rarefaction had not been done yet at the time this command is used and percentages would vary between samples.

L154. Such a wider diversity in milk. Usually this is not the norm. Not sure if due to sampling strategy or extraction protocol.

Having a precise answer to this question would require additional sampling that is beyond the scope of this work. I am also unaware of any work that collected and analyzed these three types of samples from the same cows using the same technique and would therefore have a fully comparable dataset.

L155. Proteobacteria's proportion is strangely higher than normal in the rumen. This is not usually the case.

We sampled from five sites in the rumen and processed a composite sample that included solids and liquids – this is a fairly rare approach, and perhaps this explains the discrepancy.

L157. Clarify your definition of core. From the fig legend it appears just presence in more than 60% of samples was considered as core. This is not core definition. You can say high abundant taxa. Use of core misdirects uninformed readers.

Good catch! We changed this sentence to:

*“Several OTUs which corresponded to *Christensenellaceae*, *Lachnospiraceae*, *Ruminococcaceae*, and *Ruminococcus* at the genus level, were present at a detectable abundance in each of the rumen, feces, and milk niches (Figure 1E).”*

L162. Microbial community composition is more suitable. Population refers to one species in one geographical location. You can say microbial populations but then composition after that makes it awkward.

Also, a good catch – thank you, and sorry I missed it. This has been changed throughout.

L168. data are not presented?

The data are the Chao and Shannon indices presented earlier in the sentence. We don't think it is necessary to show the boxplots as well.

L216. You are trying to link the B12 concentration in the gut with that of milk. So inclusion of dry cows was unnecessary from the beginning. The Justification here that these samples were removed because of the higher B12 concentration doesn't make sense; it makes sense to say there was no milk and these data cannot be correlated with something along your story line about B12. Unless you link it with meat concentration of B12 which again is not the focus here.

This makes sense. These sentences have been removed and the suggested justification of not having access to milk for these animals was included as follows:

“However, since milk was not available from dry animals for comparison dry animals were excluded from further comparison and statistical analysis.”

L232. ...correlated with....

The word “to” has been changed to the word “with”

L244. Community richness, or bacterial richness. It cannot be population richness

Population richness was changed to community richness.

L247, 297.....of vitamin....

The word “of” has been added in both instances.

L248. ...between communities....

“Population” has been changed to “communities.”

L266, 288, 295, 385 community richness

The word “population” has been changed to “community” in these instances.

L298. You are talking at the phylum level (Bacteroidetes) about B12 consumption. Definitely this is not the outcome of this study. Provide an accurate reference. I doubt that there are studies at the phylum level that claim all species in the given phylum are B12 consumers. But look forward

to seeing that.

A high abundance OTU that could not be identified beyond the phyla level was identified as being more prevalent in low vitamin B12 rumens. We make the point later on in the discussion that we are talking about an OTU within the Bacteroidetes phyla and not the phyla it's self in the following sentence:

“A member of the *Bacteroidetes* phyla, that could not be identified at a finer taxonomic resolution (*Prevotella* is also a member of the *Bacteroidetes* phyla), was correlated with a low concentration of vitamin B12 in the rumen.”

We do cite a reference (Wexler AG, Schofield WB, Degnan PH, Folta-Stogniew E, Barry NA, Goodman AL. Human gut Bacteroides capture vitamin B12 via cell surface-exposed lipoproteins. *eLife*. 2018;7:1324.) in our manuscript that discusses vitamin B12 consumption at the level of phyla. Degnan is a very respected researcher in the field, and are some quotes from that reference that make my point:

“In previous studies, we established that the most abundant Gram-negative bacteria in the human gut (Bacteroidetes) encode a diverse array of B12 transport systems in B12-riboswitch regulated loci, often with multiple locus architectures per genome (Degnan et al., 2014a).”

“Nearly all of the Bacteroides B12 transport loci include homologs of a hypothetical gene that is exclusively found in the Bacteroidetes phylum.”

“BtuG homologs are exclusively found among the Bacteroidetes, facilitate the acquisition of cyanocobalamin in vitro and confer a fitness advantage in gnotobiotic mice.”

“What evolutionary forces drove the Bacteroidetes, unlike other Gram-negative phyla, to incorporate an additional component into their B12 transport pathway that binds B12 with such high affinity? One possible answer lies in the gut environment, where bacteria co-exist at densities of 10^{11} cells per gram or higher (Whitman et al., 1998). Under these conditions, adaptations that increase corrinoid capture could allow cells to minimize their requirement for energetically costly vitamin biosynthetic pathways. Indeed, many human gut Bacteroidetes encode incomplete vitamin B12 biosynthesis pathways. *B. thetaiotaomicronis* missing this pathway entirely. Selection for increased corrinoid binding affinity in BtuG could conversely permit mutations that decrease the ability of BtuB to directly capture these molecules from the environment: *E. coli*, which transports B12 via BtuB and lacks any BtuG homolog, grows readily on 0.4 nM B12 (Di Girolamo et al., 1971), while *B. thetaiotaomicron* requires BtuG under these conditions (Figure 1C).”

While you are quite correct that we don't know the exact capabilities of the OTU that we are picking out here, we do find it very intriguing that this phyla was found in our study as being associated with low vitamin B12 concentrations and the literature flags this phyla as being very adapted to acquiring vitamin B12 in this type of environment. There is definitely something

worth looking at here and it is something we are actively investigating at the moment. I feel it is an interesting enough connection that it is appropriate to discuss our thinking about this finding in the discussion. I think though my wording I have made it clear that this is informed speculation.

L303. This proportion was abnormally higher than what has been observed generally in lactating cows.

Based on what study/ review? I'm not comfortable with this sentence unless the conditions of the study were very similar to ours.

L309. These types of claims require careful review of the literature and citation of appropriate references. Please either remove similar claims or provide citation to the appropriate work.

We did do a careful review of the literature; we have a reference showing that alfalfa silage was linked to increased Proteobacteria abundance (Indugu 2017). We made it clear that the link between the observations in this work and our work was speculative, but that this provides a possible explanation of the data.

L372-377. Lots of speculations here. And the last two sentences contradict each other. Please carefully review, and revise. There are more recent reviews (2018 and 2019) that have discussed this topic. Please check them out as there are not many groups who are working on milk microbiota in general and especially in dairy cows.

In spite of a thorough literature search we were unable to find a paper discussing the existence of an entero-mammary pathway in cows that is newer than 2015. We have clarified this paragraph, but we disagree that the last two sentences contradict each other.

L382. I am not comfortable with this conclusion. It requires more careful justification. Have you done OTUs tracking strategies? Apart from that, this topic wasn't the focus of this paper. Not sure why suddenly we jumped into such conclusion.

Also, a good catch - thanks. These sentences were removed from the conclusion.

L386. You are making too many strong conclusions based on a very limited and biased data. Don't forget this is an observational study not a controlled study. You have small number of animals from a wide range of DIM; therefore, concluding that diet has no effect on milk microbiota is too strong.

We also removed these sentences and pointed out that this is an observational study.

Reviewer 3:

This MS details an investigation into the factors related to the B12 content of cow's milk. The authors have investigated several possible factors, including B12 serum content, feed, and bacterial community 16S rRNA relative content of different sites along the rumen GI tract. The manuscript contains a number of correlation analyses and presents several statistical test results. The overall effect is somewhat confusing and makes one wonder if some of the results are spurious given the number of correlations attempted.

Major issues:

1. The paper lacks an underlying theoretical model for framing the investigation. In the end, it appears as though several parameters were tried. One potential starting model could be:

Bacterial taxa produce B12 -> absorption from GI tract -> serum levels -> transport to milk ducts -> output into milk (concentration vs. total output?)

Under the present state of knowledge, there are too many gaps to develop a valuable model as suggested. To explain some of these gaps:

Bacterial taxa produce B12 -> absorption from GI tract: As obtained in the current study, concentration of vitamin B₁₂ into the rumen content is only an indication of the amount of vitamin produced into the rumen because it does not take into account the volume of each ruminal fractions or the transit time. To quantify vitamin B₁₂ ruminal outflow, we need to use cows equipped with duodenal cannula and to infuse markers to quantify the duodenal flow of digesta.

Absorption from GI tract -> serum levels: Absorption of vitamin B12 happens mostly in the small intestine, as in non-ruminants. Absorption of the vitamin in the lower gastrointestinal tract is negligible because only free vitamin can be absorbed through the lower part of the gastrointestinal tract and even the efficiency of absorption of free vitamin B₁₂ is extremely low (Can. J. Anim. Sci. 2003. 83:273-278). The largest proportion of the vitamin in the lower part of the gastrointestinal tract is present within the bacteria and consequently, not available for absorption. Disappearance of vitamin B₁₂ through the small intestine could be achieved only with cows equipped with duodenal and ileal cannulas and infused with markers to estimate the intestinal flow of digesta, 45% of the amount appearing at the duodenal level disappeared before the ileal cannula (J. Dairy Sci. 2009. 92:4525-4529) but this is different from the amount of vitamin reaching blood circulation. Blood samples for plasma concentrations of vitamin B₁₂ were collected from the tail vein, after the first-pass through the liver. In dairy cow, the liver extracts 46% of the amount of vitamin B₁₂ present into the portal blood (Br. J. Nutr. 2001. 86:707-715). Consequently, plasma concentration of the vitamin is an indicator of the vitamin supply but does not allow to quantify the amount of vitamin absorbed.

Serum levels -> transport to milk ducts-> output into milk: The extraction ratio of the vitamin by the mammary gland represents only 5.5% of the arterial concentration and the mammary uptake of the vitamin is 17% greater than the amounts secreted in milk (Tierphyriol., Tierernahrg. u.

Futtermittelkde. 1985. 54:152-156). However, these values were obtained on only one cow and need to be confirmed.

Milk (concentration vs. total output?): In the present study, as milk production was highly variable among animals, and concentration is obviously influenced by the amount of milk produced, total output is more representative of the amount of vitamin secreted in milk.

This model is further complicated by the effect of feed on the bacterial community, competing bacteria taxa that utilize B12, and B12 metabolism by the cow.

Yes. This is correct.

Given that framework, do the results of the current MS allow calculation of the constants/flow of material from compartment to compartment?

No, there are too many unknown factors to allow for valuable calculations. Moreover, it is possible to quantify the amount of vitamin secreted in milk but for rumen and plasma, we have only concentrations.

What correlations/tests would be appropriate to shed some light on the flow describe above? The MS does not appear to adequately address any of these questions.

The most appropriate correlations are presented in the manuscript: rumen to plasma, plasma to milk and rumen to milk. The relationships with feces, although probably of lesser biological importance, was interesting to explore as there is no information in the literature on this subject. **The study is an exploratory study using cows under different physiological stages and feeding management in order to have a great variability.** Results from this study will add to the very limited knowledge on the relationships between the vitamin status of the animals and its microbiome. The gathered information is critical to orientating future research works on this subject.

2. The MS confuses bacterial 16S rRNA with live bacterial genera and with bacterial abundance (Bacteroides is misspelt in the MS). A 16S rRNA survey gives no indication of the relative amount of bacterial species alive at one time in a community because there are varying numbers of the rRNA operon in each bacteria. Large changes in bacterial 16s rRNA are probably indicative of big differences but one must be careful with the language used to describe these changes.

At several points in the manuscript we are referring to *Bacteroidetes* the phylum, not *Bacteroides* the genus. We have pointed out that in this particular instance the OTU could not be identified beyond the phyla level.

We agree with the stated limitations of the 16S rRNA gene survey. This is a widely known and acknowledged problem in the field (indeed it's a problem in every study that uses the 16S rRNA

gene as a proxy for bacterial abundance), and we have been as careful as possible with the language we used to compare these changes.

At this moment in time I am unaware of a way to account for differences in the rRNA operon between groups of bacteria, however, since we are using a comparative analysis between two conditions and not making conclusions about the absolute abundance of a particular genus it should be ok (or at least not worse than any other study that uses targeted amplicon sequencing).

3. The MS presents much data on changes in diversity between sites along the GI tract and vs. Milk. Why is this relevant to the B12 question? All this information could be put in supplemental material.

I think including this information contributes to the narrative of the story as well as is a full inspection of our dataset. Since vitamin B12 is synthesized in the rumen, and absorbed via the intestine, these datasets contribute enormously to our narrative.

4. Some sort of analysis of the relative importance of the relevant factors is required. For example, if one put all the factors into a Random Forest model, which ones would come out to be most influential for B12 in milk? From the MS, it appears that serum B12 is most predictive and this doesn't seem to be related to community composition? Why, then, spend so much time detailing the effect of feed/bacteria?

Because we know that feed is one of the only things that influence the concentration of vitamin B12 in milk. But feed doesn't fully explain this difference – the microbiome MUST play a role through the influence of feed. That is what we are trying to address in this paper. Serum B12 ultimately only originates from the rumen.

5. There needs to be some consideration of which bacterial taxa produce and consume B12 in the analysis. The reader thinks about which bacteria would be important to B12 while reading the introduction but this is only mentioned casually in the discussion.

I don't think we have an adequate enough list of which bacteria produce and consume for this to be included in any accurate type of analysis.

6. Why are relative phylum distributions important to the B12 question (Figure 1)? What is the point of Figure 2? Why were the genera of Figure 4 chosen and how are they related to B12?

The relative phylum distributions are not important to the vitamin B12 question, but I think they are important to the overall description of our dataset, so they were included.

The point of figure 2 was to show that there was essentially no differences in the microbiota between diet group 1 and group 2 (which were as close in composition as possible). It was also

to provide evidence that justified the elimination of the close up and dry cow samples from further analysis.

The specific genera shown in Figure 4 were chosen based on statistically significant correlations with vitamin B12 abundance based on LEFSE analysis.

Minor:

Figure 3 could best be presented as a table.

It could be. But I think the graphical presentation accurately communicates the extent of the variation in vitamin B12 between samples.

The use of multiple tests for dissimilarity is confusing. Why are there two (anosim and permanova)? If they are testing separate things, they are not presented as such. If they are testing the same thing, why are two necessary?

ANOSIM and PERMANOVA both measure if two groups are significantly different based on categorical variables (like high or low vitamin B12). Although they measure the same thing, they have strengths and weaknesses – for example PERMANOVA is sensitive to low dispersion, and so different experts prefer to use different tests. In our study, both values agreed in each instance as to if there were significant differences between samples, and it takes very little to report both values, so we did. This is not unique – I have seen the reporting of both values in several studies.

Where the correlations corrected for multiple tests (Bonferroni)?

False Discovery Rate (FDR) is more appropriate for our study – so we used that – but yes, we have corrected for multiple tests using FDR.

Why was the Chao1 estimator used instead of the Simpson index? Why is the Shannon index used? At large enough sampling (how many sequences were obtained for each community?) the Chao1 becomes the Simpson.

Chao1 measures species richness, while Simpson and Shannon measure diversity. In my work, I generally like to include Chao1 for richness and either Simpson or Shannon to for diversity. I tend to prefer Shannon since as a function calculation bacterial assemblages generally fit the prerequisite of infinite population that has been sampled randomly. Each community had a minimum of 10,000 sequences.

Where single sequences included in the analysis?

No. Single sequences were not included in the analysis.

This is problematic since there is an error rate associated with Illumina sequencing and singletons are suspect. How many singletons were observed in the data set?

We absolutely agree – and it annoys us when other authors include singletons. In this data set we did observe singletons – but since we believe that the vast majority of singletons are sequencing error, we have not included any in the analysis. In fact, we have actually eliminated all sequences that were not observed at least 20 times using the code `split.abund(fasta=stability.trim.contigs.good.unique.good.filter.fasta, count=stability.trim.contigs.good.good.count_table, cutoff=20, accnos=true)`

We have made this clear in the manuscript by adding the following sentence:

“A similarity of 97% was used as a cut-off to performed cluster process. Sequences from each sample type were analyzed together, and sequences that were not observed at least 20 times were not included in the analysis.”

What does a graph of the number of sequences in each OTU vs OTU number look like? Where is your cut off for analysis along this graph?

The graph of number of sequences in each OTU vs OTU number can be found below:

We didn't analyze anything with less than 20 sequences or anything that was found <10% of the total number of sequences in the sample.

February 6, 2020

Prof. Jennifer Ronholm
McGill University
Food Science and Agricultural Chemistry
MS1-030
21111 Lakeshore Rd
Ste-Anne-de-Bellevue, Quebec H9X3V9
Canada

Re: mSystems00107-20 (Correlations between the composition of the bovine microbiota and vitamin B12 abundance)

Dear Prof. Jennifer Ronholm:

Thank you for the thoughtful and detailed response to the reviewers concerns.

Your manuscript has been accepted, and I am forwarding it to the ASM Journals Department for publication. For your reference, ASM Journals' address is given below. Before it can be scheduled for publication, your manuscript will be checked by the mSystems senior production editor, Ellie Ghatineh, to make sure that all elements meet the technical requirements for publication. She will contact you if anything needs to be revised before copyediting and production can begin. Otherwise, you will be notified when your proofs are ready to be viewed.

Sincerely,

Jack Gilbert
Editor, mSystems

Journals Department
Table S5: Accept
Supplemental Material: Accept
Table S1: Accept
Table S4: Accept
Table S8: Accept
Table S2: Accept
Table S3: Accept
Table S6: Accept
Table S7: Accept